# Decoupling Dynamical Richness from Representation Learning: Towards Practical Measurement

**Yoonsoo Nam[1], Nayara Fonseca[1], Seok Hyeong Lee[2], Chris Mingard[1], Niclas Goring[1], Ouns El Harzli[1], Abdurrahman Hadi Erturk[1], Soufiane Hayou[3], Ard A. Louis[1]**
[1]Oxford University, [2]Seoul National University, [3]Johns Hopkins University

## Abstract

Dynamic feature transformation (the rich regime) does not always align with predictive performance (better representation), yet accuracy is often used as a proxy for richness, limiting analysis of their relationship. We propose a computationally efficient, performance-independent metric of richness grounded in the low-rank bias of rich dynamics, which recovers neural collapse as a special case. The metric is empirically more stable than existing alternatives and captures known lazy-to-rich transitions (e.g., grokking) without relying on accuracy. We further use it to examine how training factors (e.g., learning rate) relate to richness, confirming recognized assumptions and highlighting new observations (e.g., batch normalization promotes rich dynamics). An eigendecomposition-based visualization is also introduced to support interpretability, together providing a diagnostic tool for studying the relationship between training factors, dynamics, and representations.

## 1 Introduction

In machine learning, feature learning is often viewed through two complementary perspectives: improvement of representations and non-linear training dynamics. The *representation perspective* emphasizes feature quality — how well it supports downstream tasks like classification and promotes generalization (Bengio et al., 2013). The *dynamics perspective* — often called rich regime in *rich versus lazy training* (Chizat et al., 2019) — concerns the dynamic transformation of features beyond linear models. While dynamical richness frequently correlates with representation usefulness, rich dynamics reflect a preference (inductive bias) toward certain solutions, without necessarily benefiting all tasks (Geiger et al., 2020; Göring et al., 2025). For example, dynamic feature learning can impair the performance on an image classification task (Fig. 1).

The representation perspective of feature learning — central to deep learning's success — remains poorly understood. However, its dynamic aspects, often shaped in practice by initialization and optimization, are well understood in the context of layerwise linear models (e.g., linear networks). Greedy dynamics with low-rank biases (Saxe et al., 2014; Mixon et al., 2020) explains neural collapse (Papyan et al., 2020) in vision tasks; delayed saturation dynamics (Nam et al., 2024) has been linked to emergence (Brown et al., 2020) and scaling laws (Kaplan et al., 2020) in large language models; and initialization-dependent dynamics (Kumar et al., 2024; Kunin et al., 2024; Lyu et al., 2024) offers insights into grokking (Power et al., 2022) in arithmetic tasks. See Nam et al. (2025) for a comprehensive overview.

To better understand the link between rich dynamics and representation improvement, we need independent metrics for dynamics and representations. However, prior dynamics metrics, while insightful, are not optimized for practical richness measurement (Section 4.1). In this paper, we propose a dynamical richness metric that compares the activations before and after the last layer. The metric (1) generalizes neural collapse, (2) is computationally cheap, (3) is performance-independent, and (4) operates in the function space. We demonstrate that our metric empirically tracks dynamical richness without referencing performance, and assess its robustness against existing metrics. Finally, we experiment our metric, with complementary visualization, across various setups, highlighting its potential as a tool for uncovering new empirical insights.

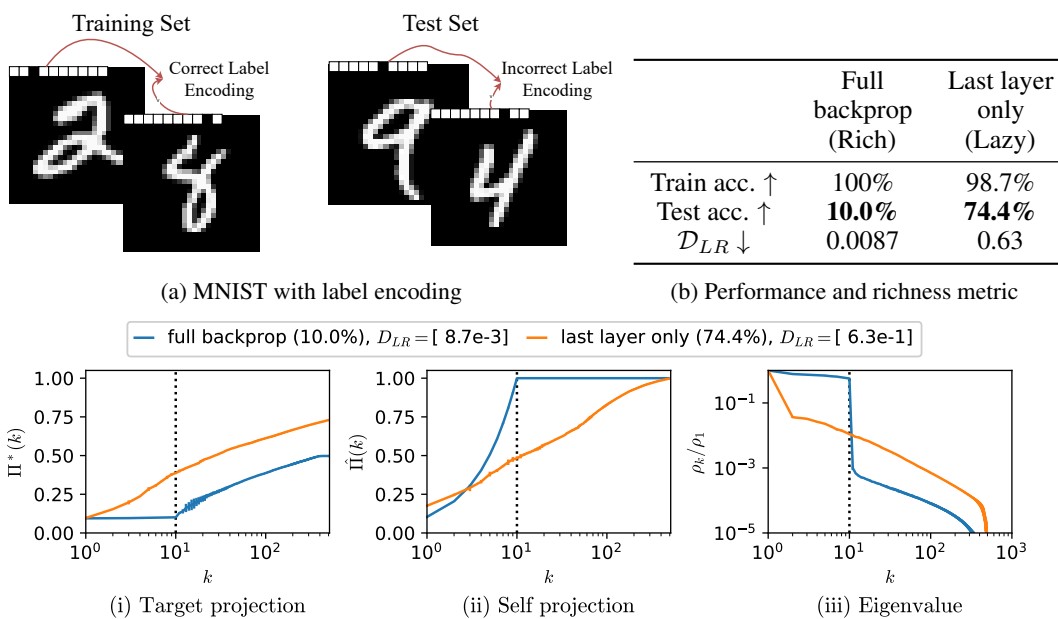

(a) MNIST with label encoding

(b) Performance and richness metric

| | Full backprop (Rich) | Last layer only (Lazy) |
|---|---|---|
| Train acc. ↑ | 100% | 98.7% |
| Test acc. ↑ | **10.0%** | **74.4%** |
| $\mathcal{D}_{LR}$ ↓ | 0.0087 | 0.63 |

full backprop (10.0%), $D_{LR} = [\,8.7e\text{-}3\,]$    last layer only (74.4%), $D_{LR} = [\,6.3e\text{-}1\,]$

(i) Target projection     (ii) Self projection     (iii) Eigenvalue

(c) Visualization of representation quality (i) and our metric (ii,iii) using the eigendecomposition

Figure 1: **Rich dynamics $\neq$ better generalization.** We trained a 4-layer MLP on label-encoded MNIST. **(a):** The first 10 pixels are encoded with true labels in training and random labels in testing; both the encoding and the image serve as valid features for training. **(b):** A full backpropagation model *(rich)* biases toward the encodings and generalizes poorly, while a last-layer-only-trained model *(lazy)* relies on the full image and generalizes better. Our low-rank-based metric $\mathcal{D}_{LR}$ (Eq. (4)) quantifies the dynamical richness ($\mathcal{D}_{LR} \in [0, 1]$ where 0 is richest) independent of the performance. **(c):** Complementary visualization method (Eq. (5)): (i) Cumulative contribution of last-layer features in expressing the target function — top 10 features are irrelevant in the rich model. (ii) Contribution to the learned function — the rich model uses only the top 10 features, while the lazy model uses all. (iii) Relative feature norms — rich model concentrates on the top 10; lazy model decays more gradually. Test accuracies and $\mathcal{D}_{LR}$ values are shown in parentheses and square brackets, respectively. See Section 5.1 for details, and Section C for background and motivation.

**Contributions:**

- We introduce the Minimum projection operator $\mathcal{T}_{MP}$ (Definition 1) and define the dynamical low-rank measure $\mathcal{D}_{LR}$ (Eq. (4)), a lightweight and performance-independent metric.

- We show that $\mathcal{D}_{LR}$ reduces to neural collapse as a special case, connecting our formulation to a well-studied phenomenon while extending it to settings without labels.

- We empirically confirm that $\mathcal{D}_{LR}$ captures known lazy-to-rich transitions such as grokking and target downscaling (Table 2), aligning with theoretical expectations while giving a direct evaluation without referencing performance. We also empirically assess its robustness against alternatives.

- We demonstrate the utility of our metric by showing how various training factors (e.g., architecture and learning rate) relate to richness and performance (Table 3). Our metric explicitly confirms prior assumptions and uncovers new observations, such as batch normalization shifting VGG-16 on CIFAR-100 from lazy to rich dynamics. We further adapt eigendecomposition to visualize the metric for improved interpretability.

## 2 SETUP AND BACKGROUND

For analytical tractability, we use MSE loss and supervised settings where target function entries are orthogonal and isotropic (e.g., balanced $C$-way classification or regression with scalar output).

**Notations.** For the input space $\mathcal{X}$, we train on $n$ samples from $q$, the underlying true distribution on the input space $\mathcal{X}$. The width of the last layer (or the linear model) is $p$. The dimension of the output function (and the output of the neural network) is $C$. Denote $\hat{f}$ and $f^*$ as the learned function (at any point in training) and the target function, respectively. We use bra-ket notation $\langle f|g\rangle := \mathbf{E}_{x\sim q}[f(x)g(x)]$ to mean the inner product over the input distribution (Section B). See Section A for a **glossary**.

**Empirical details.** Full experimental details, link to the source code, and statistical significance for the tables are provided in Section H. All error bars represent one standard deviation.

**Feature kernel operator.** We define the *feature map* $\Phi : \mathcal{X} \to \mathbb{R}^p$ as the map from the input to the post-activations of the penultimate layer. For $1 \le k \le p$, denote the $k^{\text{th}}$ *feature* as the $k^{\text{th}}$ entry $\Phi_k(x)$ of $\Phi(x) = [\Phi_1(x), \dots, \Phi_p(x)]$. We define the *feature kernel operator* $\mathcal{T} : L^2(\mathcal{X}) \to L^2(\mathcal{X})$, generalization of self-correlation matrix in the function space, as

$$\mathcal{T}[f](x') := \mathbf{E}_{x\sim q}\left[\sum_{k=1}^{p}\Phi_k(x)\Phi_k(x')f(x)\right], \qquad \mathcal{T} = \sum_{k=1}^{p}|\Phi_k\rangle\langle\Phi_k|, \tag{1}$$

where $L^2(\mathcal{X})$ is the Hilbert space of square-integrable functions on $\mathcal{X}$ (with distribution $q$). The second expression is $\mathcal{T}$ in bra-ket notation (Section B). Using Mercer's theorem (Mercer, 1909), we can decompose the operator into eigenfunctions and eigenvalues

$$\mathcal{T} := \sum_{k=1}^{p}\rho_k|e_k\rangle\langle e_k|, \qquad \mathcal{T}[e_k] = \rho_k e_k, \qquad \langle e_k|e_l\rangle = \delta_{kl}, \tag{2}$$

where $\rho_k \in \mathbb{R}$ is a non-negative eigenvalue and $e_k : \mathcal{X} \to \mathbb{R}$ is the orthonormal eigenfunction. The operator $\mathcal{T}$ and the eigenfunctions $e_k$ play a key role in kernel regression with feature map $\Phi$ (i.e. $K(x, x') = \Phi(x)^T\Phi(x')$). The linear model (kernel) has inductive bias toward eigenfunctions corresponding to larger eigenvalues (Bordelon et al., 2020; Jacot et al., 2020; Spigler et al., 2020; Canatar et al., 2021; Cui et al., 2021; El Harzli et al., 2024), and generalizes better when eigenfunctions with larger corresponding eigenvalues also better describe the target function (larger $\langle e_k|f^*\rangle$) — also known as the task-model alignment (Bordelon et al., 2020). See Section C for a gentle introduction.

**Rich dynamics.** Chizat et al. (2019) introduced rich dynamics as deviations from the exponential saturation observed in linear models. Subsequent works (Kunin et al., 2024; Dominé et al., 2025; Nam et al., 2025) identified these dynamics as sigmoidal saturation or amplifying dynamics — arising from gradient descent in layerwise models — and connected them to phenomena like neural collapse (Mixon et al., 2020), feature emergence (Nam et al., 2024), and grokking (Kunin et al., 2024). See Nam et al. (2025) for a comprehensive overview.

**Low-rank bias and neural collapse.** Low-rank hidden representations naturally emerge in the rich regime, as shown by studies across linear networks (Saxe et al., 2014; Ji & Telgarsky, 2019; Arora et al., 2019a; Lampinen & Ganguli, 2019; Gidel et al., 2019; Tarmoun et al., 2021), unconstrained feature models (Mixon et al., 2020; Fang et al., 2021), and matrix factorization (Arora et al., 2019b; Li et al., 2020). These works find that gradient dynamics decouple into a minimal number of modes, governed by the rank of the input-output correlation matrix (often that of the output $C$), causing gradients to concentrate on those modes and produce low-rank representations. In practice, the low-rank bias is observed as neural collapse (Papyan et al., 2020), which consists of four conditions NC1-4 stated in Section D. Although often associated with improved representations, neural collapse does not consistently imply better generalization (Zhu et al., 2021; Nguyen et al., 2022; Hui et al., 2022; Su et al., 2023), remaining largely a dynamical phenomenon. See Kothapalli et al. (2022) for an overview.

**Change in NTK as measure of rich dynamics.** Beyond performance, changes in the Neural tangent kernel (NTK, Jacot et al. (2018)) before and after training are among the most commonly used proxies for rich dynamics (Chizat et al., 2019; Atanasov et al., 2021; Kumar et al., 2024). Although theoretically well-founded, NTK-based measures scale with the total number of parameters and are computationally infeasible even for moderate vision models. For this reason, we omit NTK from our analysis and we focus only on the feature kernel of the last layer (Eq. (1)).

## 3 THE RICHNESS MEASURE $\mathcal{D}_{LR}$ (MINIMUM PROJECTION ALIGNMENT)

We introduce a dynamical richness measure that exploits the low-rank bias of rich dynamics and show that it reduces to neural collapse as a special case.

### 3.1 LOW RANK BIAS AS RICHNESS MEASURE

We define the *learned function space* $\hat{\mathcal{H}} = \text{span}\{\hat{f}_1, \ldots, \hat{f}_C\}$ where $\hat{f}_k : \mathcal{X} \to \mathbb{R}$ is the $k^{\text{th}}$ entry of the network's learned function (at any given time). In an ideal rich dynamics scenario, only the minimal number of features are learned throughout the dynamics and are sufficient to express (linearly span) the learned function space. Leveraging this idea, we define the minimal projection operator $\mathcal{T}_{MP}$.

**Definition 1** (Minimum Projection (MP) operator). *For a neural network with learned function $\hat{f}$ and features $\Phi(x)$, the corresponding $\mathcal{T}$ is an MP-operator $\mathcal{T}_{MP}$ if it can be expressed as*

$$\mathcal{T}_{MP}[u] = a_1 \langle \mathbf{1} | u \rangle \mathbf{1} + a_2 P_{\hat{\mathcal{H}}}(u) \quad \text{for all } u \in L^2(\mathcal{X}), \tag{3}$$

*where $a_1, a_2 > 0$, $\mathbf{1} \in L^2(\mathcal{X})$ is a constant function, and $P_{\hat{\mathcal{H}}} : L^2(\mathcal{X}) \to L^2(\mathcal{X})$ is the orthogonal projection in $L^2(\mathcal{X})$ onto $\hat{\mathcal{H}}$.*

Ignoring the constant function (setting $a_1 = 0$), whose discussion is deferred to Section D, the $\mathcal{T}_{MP}$ is (up to a constant scale) a projection operator $P_{\hat{\mathcal{H}}}$ that removes all components orthogonal to $\hat{\mathcal{H}}$. If $\mathcal{T}$ is $\mathcal{T}_{MP}$, the last-layer features span only a $C$-dimensional space that matches the learned function space, reflecting the low-rank structure characteristic of rich dynamics. We thus define the *low rank measure* $\mathcal{D}_{LR}$ as the similarity between the $\mathcal{T}$ — defined by the current features — and the MP-operator $\mathcal{T}_{MP}$ — defined by the current learned function — as a metric for richness:

$$\boxed{\mathcal{D}_{LR} := 1 - CKA(\mathcal{T}, \mathcal{T}_{MP}),} \tag{4}$$

where $CKA$ is the centered kernel alignment (Kornblith et al., 2019) with bounded value in $[0, 1]$. Because $CKA$ is normalized and uses centered (zero-mean) alignment, the metric remains consistent for any $a_1$ and $a_2$ (Section B.3). We subtract the $CKA$ measure from 1 so that lower values indicate richer dynamics, consistent with more widely used metrics of richness. The novelty of our metric $\mathcal{D}_{LR}$ lies in defining the minimum projection operator $\mathcal{T}_{MP}$ and comparing it to $T$ to **quantify dynamical richness**, not in using well-established CKA. Notably, CKA is primarily used to compare the NTK before and after training (Chizat et al., 2019; Kumar et al., 2024; Baratin et al., 2021).

**Intuition.** Our metric (Eq. (4)) compares the activations **before** ($\mathcal{T}$, at most rank $p$) and **after** ($\mathcal{T}_{MP}$, rank $C$) the last layer. In rich dynamics, low-rank bias dictates that only the minimal necessary $C$ **out of** $p$ features are learned and used before the final layer ($\mathcal{D}_{LR} = 0$). In this case, the final layer performs no additional processing because earlier layers have already completed the task. Conversely, an excess of features indicates a bottleneck — such as limited expressivity in earlier layers — that prevents the full manifestation of low-rank bias.

**Time complexity.** The algorithm is highly efficient. For $n$ samples (from either train or test), it requires $n$ forward passes to record activations before and after the final layer, producing matrices of size $n \times p$ and $n \times C$, where $p$ is the **last-layer width** and $C$ the number of classes. $\mathcal{D}_{LR}$ is then computed in $\mathcal{O}(npC)$. Since typically $n \gg p \gg C$, setting $n \approx \mathcal{O}(p)$ suffices, reducing complexity to $\mathcal{O}(p^2 C)$. In standard models with $p \approx 10^3$, this is far cheaper than NTK-based methods, which scale quadratically with the **total number of parameters**. See Section E for details.

### 3.2 CONNECTION TO NEURAL COLLAPSE

Suppose that the empirical distribution coincides with the true distribution and the neural network perfectly classifies all labels with one-hot vectors. We show that if $\mathcal{T}$ is an MP-operator, then the NC1 and NC2 conditions (Section D) of neural collapse hold. Let dagger † denote pseudo inverse, $A_i$ the set of datapoints with class label $i$, so the $i^{\text{th}}$ learned function $\hat{f}_i$ coincides with the one-hot indicator function for $A_i$. Following Papyan et al. (2020), we define the $i^{\text{th}}$ class mean vector as $\mu_i := \mathbf{E}_{x \in A_i}[\Phi(x)]$ and the global mean vector as $\bar{\mu} := C^{-1} \sum_{i=1}^{C} \mu_i$.

**Proposition 1.** *If $\mathcal{T}$ is an MP-operator, then the **NC1** condition (collapse of within-class variability)* $\Sigma_b^\dagger \Sigma_W = 0$ *holds, where inter-class covariance matrix $\Sigma_b$ and intra-class covariance matrix $\Sigma_W$ are*

$$\Sigma_b = \sum_{j=1}^{C}(\mu_j - \bar{\mu})(\mu_j - \bar{\mu})^T \quad and \quad \Sigma_W = \sum_{i=1}^{C} \mathbf{E}_{x \in A_i}\left[ (\Phi(x) - \mu_i)(\Phi(x) - \mu_i)^T \right].$$

**Proposition 2.** *If $\mathcal{T}$ is an MP-operator, then the **NC2** condition (convergence of features to a simplex equiangular tight frame) holds:*

$$(\mu_i - \bar{\mu})^T (\mu_j - \bar{\mu}) \propto \delta_{ij} - \frac{1}{C}.$$

See Section D for proofs and further discussions.

Although the ideal criteria of $\mathcal{T}$ is $\mathcal{T}_{MP}$ implies neural collapse criteria as a special case, they differ in general: we measure how well **features (random variables or functions)** express the **learned function**, while neural collapse concerns how training **feature vectors** represent **class mean vectors**. Based on the function space, our measure extends to test data, enables feature quality assessment (Eq. (5)), and is more empirically robust (Tables 1 and 2), extending beyond neural collapse (see Section D).

## 4 EXPERIMENTS

Here, we share our empirical results. We first compare our metric with existing metrics of richness, then we confirm that our measure empirically tracks known lazy-to-rich transitions. We further conduct experiments on various training setups to explore the relationship between training factors, rich dynamics, and performance (better representations).

### 4.1 COMPARISON TO PRIOR MEASURES OF RICH DYNAMICS

Three alternative metrics for richness measures are: (1) similarity to the initial kernel, $\mathcal{S}_{init} = CKA(K_{\text{init}}, K_{\text{learned}}) \in [0, 1]$ (Yang & Hu, 2021), (2) the parameter norm $\|\theta\|_F^2$ (Lyu et al., 2024), and (3) class separation from neural collapse, $NC1 = \text{Tr}(\Sigma_b^\dagger \Sigma_W)$ (Papyan et al., 2020; Stevens et al., 2002; He & Su, 2023; Xu & Liu, 2023; Súkeník et al., 2024), where $\Sigma_b$ and $\Sigma_W$ are the inter- and intra-class covariances (Proposition 1). Prior metrics depend on the initial kernel, parameter norms, or class labels, which can constrain their use as independent measures of richness.

Table 1: Richness for weight decay

| Epoch | 0 (init) | 200 |
|---|---|---|
| Train Acc.↑ | 10.0% | 10.2% |
| Test Acc.↑ | 9.85% | 10.0% |
| $\mathcal{D}_{LR}$↓ | **0.59** | **1.0** |
| $\mathcal{S}_{init}$↓ | 1.0 | 0.20 |
| $\|\theta\|_F^2$↓ | $3.1 \cdot 10^3$ | $2.2 \cdot 10^{-5}$ |
| $NC1$↓ | $1.2 \cdot 10^5$ | $7.5 \cdot 10^{-14}$ |

Table 2: Richness metrics for target downscaling

| $\alpha$ | $2 \cdot 10^{-1}$ | $2 \cdot 10^0$ | $2 \cdot 10^1$ |
|---|---|---|---|
| Train Acc.↑ | 100 % | 100% | 100% |
| Test Acc.↑ | 92.7% | 92.4% | 88.3% |
| $\mathcal{D}_{LR}$↓ | $4.9 \cdot 10^{-2}$ | $1.1 \cdot 10^{-1}$ | $\mathbf{5.6 \cdot 10^{-1}}$ |
| $\mathcal{S}_{init}$↓ | $6.8 \cdot 10^{-2}$ | $4.1 \cdot 10^{-2}$ | $5.2 \cdot 10^{-2}$ |
| $\|\theta\|_F^2$↓ | $3.4 \cdot 10^3$ | $3.2 \cdot 10^3$ | $3.2 \cdot 10^3$ |
| $NC1$↓ | $2.3 \cdot 10^4$ | $3.2 \cdot 10^3$ | $8.1 \cdot 10^2$ |

Table 1 shows an extreme case of training an MLP on MNIST with large weight decay and a negligible learning rate. Here, the dynamics are dominated by $L^2$ weight decay, with little meaningful learning. While existing metrics can sometimes misinterpret this as rich behavior (smaller after training), our span-based measure correctly identifies the lack of dynamical richness (bigger after training).

Table 2 presents a more practical setting, where we tune laziness via target downscaling (i.e., $y \rightarrow y/\alpha$). Prior works (Chizat et al., 2019; Geiger et al., 2020) show that scaling targets by a factor $\alpha$ induces lazier training where larger $\alpha$ implies greater laziness. A good metric should capture this. Our measure **aligns with** $\alpha$, while all other measures misalign with laziness. By not relying on initial kernel, weight norm, or labels, our metric shows greater robustness in this setup. See Fig. 11 in Section F for visualization of the dynamics in Table 2.

**Kernel distance from initialization ($\mathcal{S}_{init}$).** Kernel deviation (Yang & Hu, 2021), albeit often the comparison of NTKs, is a common measure of rich dynamics (Chizat et al., 2019). However, it measures the deviation from the *initial kernel*, not *how* the kernel changed (e.g. weight decay alone changes the kernel in Table 1). While theoretically appealing for near-lazy models, the metric can be less accurate in deep, rich-training regimes where most practical training occurs.

**Parameter norm ($\|\boldsymbol{\theta}\|_F^2$).** Smaller parameter norms often correlate with rich dynamics in practice, similar to how the authors of Lyu et al. (2024) used it in their study of grokking. However, small weights promote rich dynamics (Kumar et al., 2024; Atanasov et al., 2024; Nam et al., 2025; Saxe et al., 2014; 2019), not the other way around. In fact, rich dynamics can occur for larger initialization (Braun et al., 2022; Dominé et al., 2025) and depends on broader factors such as layer imbalances (Kunin et al., 2024; Dominé et al., 2025; Nam et al., 2025).

**Neural collapse measure ($NC1$).** Separation-based metrics measure how training samples deviate relative to class boundaries (Papyan et al., 2020; Stevens et al., 2002; He & Su, 2023; Xu & Liu, 2023; Súkeník et al., 2024). While most similar to our metric, they are unbounded, sensitive to output scaling, and can be empirically ill-conditioned. As shown in Table 1, these issues can lead to dramatic value shifts and hinder interpretability.

**Low-rank measure ($\mathcal{D}_{LR}$).** Our proposed metric evaluates the alignment between features and the learned function, achieving its optimum when they span the same space isotropically (i.e., a scaled projection operator). Our metric exploits the low-rank bias of rich dynamics through alignment and is normalized between $[0, 1]$. Crucially, it does so without relying on class labels, accuracy, or the initial kernel, making the measure more appealing as an independent measure of richness.

## 4.2 EMPIRICAL FINDINGS REGARDING TRAINING FACTORS

Table 3: Richness measure and performance on various setups

| Task | Architecture | Condition | Train acc. | Test acc. | $\mathcal{D}_{LR}$ | Figure |
|---|---|---|---|---|---|---|
| Mod 97 | 2-layer transformer | Step 200 (before grokking) | 100% | 5.2% | 0.51 | Fig. 12 |
| | | Step 3000 (after grokking) | 100% | 99.8% | 0.11 | |
| CIFAR-100 | ResNet18 | learning rate = 0.005 | 100% | 66.3% | 0.053 | Fig. 4 |
| | | learning rate = 0.05 | 100% | 78.3% | 0.025 | |
| | | learning rate = 0.2 | 100% | 74.5% | 0.039 | |
| CIFAR-10 | ResNet18 | weight decay = 0 | 100% | 93.5% | 0.05 | Fig. 13 |
| | | weight decay = $10^{-4}$ | 100% | 94.1% | 0.015 | |
| | | weight decay = $10^{-3}$ | 100% | 94.8% | 0.003 | |
| CIFAR-10 | ResNet18 MLP | This experiment compares architectures only. | 100% | 94.8% | 0.026 | Fig. 14 |
| | | | 99.8% | 55.4% | 0.48 | |
| CIFAR-10 | ResNet18 | no label shuffling | 100% | 95.0% | 0.031 | Fig. 15 |
| | | 10% label shuffling | 100% | 66.1% | 0.042 | |
| | | full label shuffling | 100% | 9.5% | 0.034 | |
| MNIST | CNN | full backpropagation | 100% | 99.1% | 0.043 | Fig. 16 |
| | | last layer only training | 99.7% | 96.8% | 0.51 | |
| CIFAR-100 | VGG-16 | without batch normalization | 99.5% | 21.7% | 0.66 | Fig. 3 |
| | | with batch normalization | 100% | 72.0% | 0.073 | |

Table 3 demonstrates the practical usefulness of our performance-independent metric, summarizing the correlation between training factor, performance, and richness in various setups. The visualization of most experiments is provided in Section F.

The first row empirically confirms that our metric is a richness metric, explicitly capturing the known lazy-to-rich transition of grokking (Kumar et al., 2024; Kunin et al., 2024; Lyu et al., 2024; Nam et al., 2025).

Rows two to four confirm the assumptions that the optimal learning rate ($2^{nd}$ row), the weight decay ($3^{rd}$ row), and the architecture ($4^{th}$ row) achieve high performance **through rich dynamics** (Ginsburg

et al., 2018; Liu et al., 2022; He et al., 2016). Indeed, our metric makes this link **explicit** by having a smallest value for the optimal setting. Although such relationships are broadly recognized, they are typically only addressed implicitly.

The fifth and sixth rows, like Fig. 1, demonstrate that rich dynamics do not strictly correlate with performance. ResNet18 on CIFAR-10 retains rich dynamics even when the labels are shuffled, and convolutional neural network (CNN) on MNIST can achieve similar performance through both lazy and rich dynamics.

The last row reveals a **new observation**: VGG-16 on CIFAR-100 is lazy without batch normalization but rich with it, accompanied by a significant performance gap. While the performance effect of batch normalization is empirically established, its underlying role remains debated. Our metric helps clarify this by reframing generalization in terms of the more tractable problem of rich dynamics.

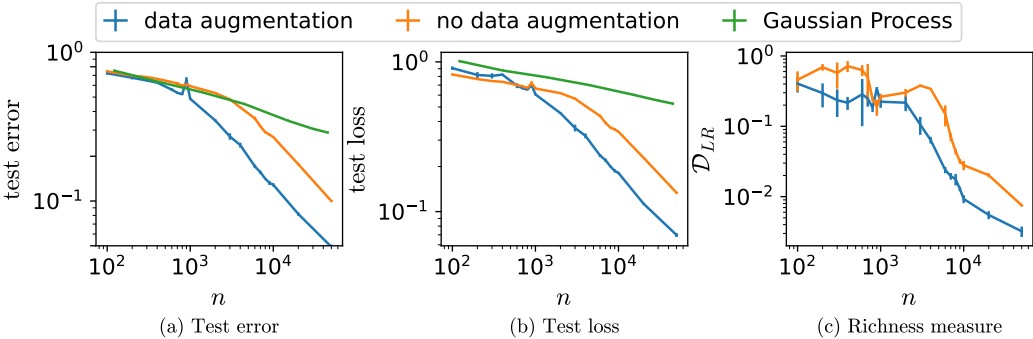

(a) Test error         (b) Test loss         (c) Richness measure

Figure 2: **Learning curve and feature learning metric**. **(a):** Learning curves of ResNet18 on CIFAR-10. Both error (a) and loss (b) learning curves show a transition to a faster-decaying power law with additional data near $n \approx 10^3$, correlating with the shift in decay of the richness measure $\mathcal{D}_{LR}$ in (c). This agrees with theoretical study on phase transition (Rubin et al., 2024) that a sufficiently large number of data points is critical for rich dynamics — a promising observation toward better understanding feature learning dynamics. A linear model (Gaussian process) was plotted in (a,b) to highlight the transition into faster-decaying learning curve.

## 5   VISUALIZATION METHODS

While metrics provide quantitative summaries, component-wise visualization aids interpretation. For instance, how significant is the contrast between VGG-16 trained with versus without batch normalization (Fig. 3)? Or why does a small change in $\mathcal{D}_{LR}$ correspond to large performance differences under varying learning rates (Fig. 4)? Does the CIFAR-10 MLP begin in a lazy regime, or transition into it during training (Fig. 14 in Section F)? To address such questions, we introduce a complementary visualization based on widely used eigendecomposition of kernel (Eq. (1)).

### 5.1   VISUALIZATION THROUGH DECOMPOSED FEATURES

Bengio et al. (2013) described feature learning as the process of learning better representations for a downstream model, such as the classifier. Building on this idea, we view the earlier layers, represented by the feature map $\Phi : \mathcal{X} \to \mathbb{R}^p$, as providing improved features for the final linear layer. Combined with our $\mathcal{T}$-dependent richness metric, this motivates a visualization method that extends the linear model analysis of $\mathcal{T}$. See Section C for related works and a gentle overview.

Our visualization quantifies three aspects (cumulative quality $\Pi^*(k)$, cumulative utilization $\hat{\Pi}(k)$, and relative eigenvalue $\rho_k/\rho_1$) of the last-layer features through the eigenfunctions $e_k$ of $\mathcal{T}$:

$$(i) : \Pi^*(k) = \sum_{j=1}^{k} \frac{\langle e_k | P_{\mathcal{H}^*}[e_k] \rangle^2}{\dim(\mathcal{H}^*)}, \qquad (ii) : \hat{\Pi}(k) = \sum_{j=1}^{k} \frac{\langle e_k | P_{\hat{\mathcal{H}}}[e_k] \rangle^2}{\dim(\hat{\mathcal{H}})}, \qquad (iii) : \rho_k/\rho_1, \quad (5)$$

where $\mathcal{H}^* = \mathrm{span}\{f_1^*, \ldots, f_C^*\}$ and $\hat{\mathcal{H}} = \mathrm{span}\{\hat{f}_1, \ldots, \hat{f}_C\}$ are the target and learned function spaces, and $P_{\mathcal{H}} : L^2(\mathcal{X}) \to \mathcal{H}$ is the projection operator onto the space $\mathcal{H}$. All cumulative measures lie in $[0, 1]$, reflecting how well the top $k$ features span the respective space. Notably, $\Pi^*(k)$ is the cumulative power in Canatar & Pehlevan (2022), quantifying the contribution of the top features in expressing the target.

These three measures together capture complementary views of feature learning. The cumulative quality ($\Pi^*(k)$) reflects how well the features align with the task. The utilization ($\hat{\Pi}(k)$) indicates how many features are used by the final layer, while the relative eigenvalues ($\rho_k/\rho_1$) show their relative magnitudes or importance. The latter two ($\hat{\Pi}$ and $\rho_k/\rho_1$) decompose the deviation from $\mathcal{D}_{LR} = 0$ condition by visualizing how many features are used and how many are significant (non-negligible).

## 5.2 CALCULATING THE VISUALIZATION METRICS

To calculate the eigenvalues and eigenfunctions (and thus the quality, utilization, and intensity) for a feature map $\Phi : \mathcal{X} \to \mathbb{R}^p$, we need the true input distribution $q$, which is inaccessible. However, we can use Nyström method (Baker & Taylor, 1979; Williams & Rasmussen, 2006) for a sufficiently large sample size of $n > p$ to approximate the eigenfunctions and eigenvalues of interest. We can create a $p \times p$ empirical self-covariance matrix $\Sigma$ from $n$ samples of $\Phi(x)$. By diagonalizing $\Sigma = USU^T$, we can obtain the empirical eigenvectors and eigenvalues, which can be used to approximate the true eigenvalues and **eigenfunctions** (Algorithm 1):

$$\rho_k \approx s_k, \qquad e_k(x) \approx \Phi(x)^T u_k / \sqrt{s_k}. \tag{6}$$

---

**Algorithm 1** Empirical eigenfunctions and eigenvalues

1:   $\Phi, D_{tr}$ (Prepare a feature map $\Phi$ and $D_{tr}$: $n$ samples of training set)
2:   $\phi \leftarrow \Phi(D_{tr})$ (forward transform the input samples to feature vectors)
3:   $U, S, U^T \leftarrow SVD(\phi\phi^T)$ (perform singular value decomposition)
4:   $[u_1, u_2, \ldots, u_p] \leftarrow U$ (find the column vectors of $U$)
5:   **for** $k$ in $1, 2, \ldots, p$ **do**
6:      $\rho_k \leftarrow s_k$ (approximate eigenvalues)
7:      $e_k(x) \leftarrow \Phi(x)^T u_k / \sqrt{s_k}$ (approximate eigenfunctions)
8:   **end for**

---

**Functions not vectors.** Note that eigenfunctions $e_k's$ are defined **beyond the training samples** used to compute $\phi$ and differ from the eigenvectors of the empirical self-covariance matrix. This enables us to evaluate feature quality — via inner products with the target function — on the **test set** rather than the training set, a capability not available for empirical eigenvectors. For validity of approximation and dependency on sample size, see Section J. As eigendecomposition arises naturally in study of kernels, similar visualizations have appeared in prior works (e.g., (Canatar et al., 2021)); see Section G for details.

## 5.3 VISUALIZATION RESULTS

**Interpreting visualization** In perfectly rich dynamics (e.g., blue line in Fig. 3), $\hat{\Pi}(C) = 1$ in (ii), showing that only the first $C$ features are used by the last layer. Additionally, $\rho_2 \approx \rho_C$ and $\rho_C \gg \rho_{C+1}$ in (iii), indicating that the features have only $C$ significant dimensions. The first eigenvalue, corresponding to a constant function, is ignored because of the centering in CKA (see Section D). If the performance is high, $\Pi^*(C) \approx 1$ in (i), showing that first $C$ features well express the target. For the lazy regime, more eigenfunction will be used (slower capping in (ii)) and the feature occupy higher dimensions (slower decay in (iii)).

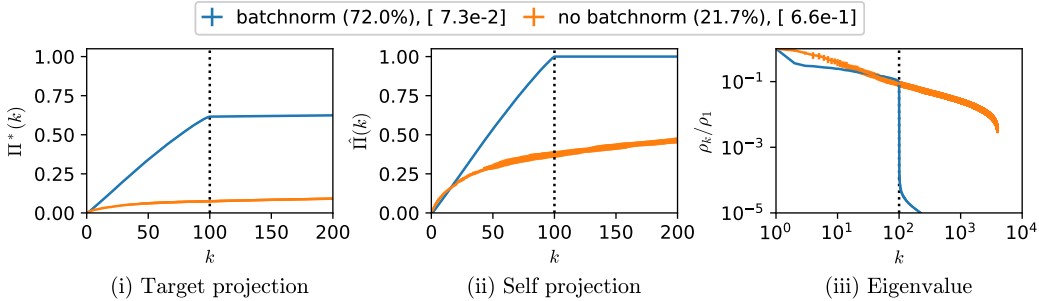

Figure 3: **Visualization of VGG16 on CIFAR-100 with and without batch normalization.** We visualize the last row of Table 3, where batch normalization shifts the model from the lazy to the rich regime. The eigenvalue distribution (iii) highlights this difference: with batch normalization, only 100 features are significant, whereas without it the eigenvalues decay slowly.

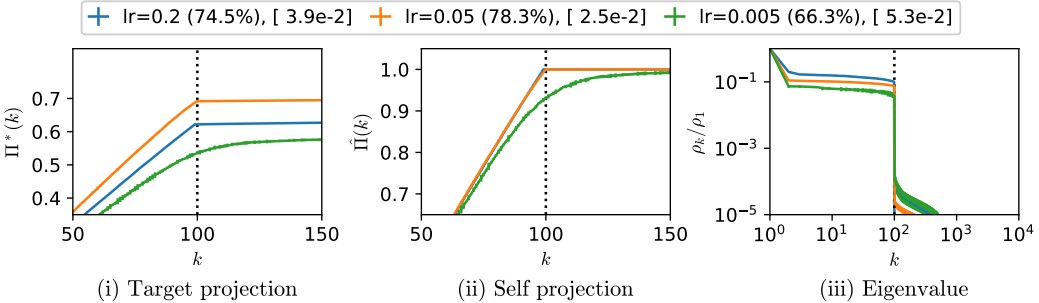

Figure 4: **Visualization on the role of learning rate.** We visualize the $2^{nd}$ row of Table 3 where the learning rates are varied (up to training instability) for ResNet18 on CIFAR-100. The second column (ii) shows that smallest learning rate uses significantly more eigenfunctions (features), while other models uses minimal 100 eigenfunctions, indicating a lazier dynamics.

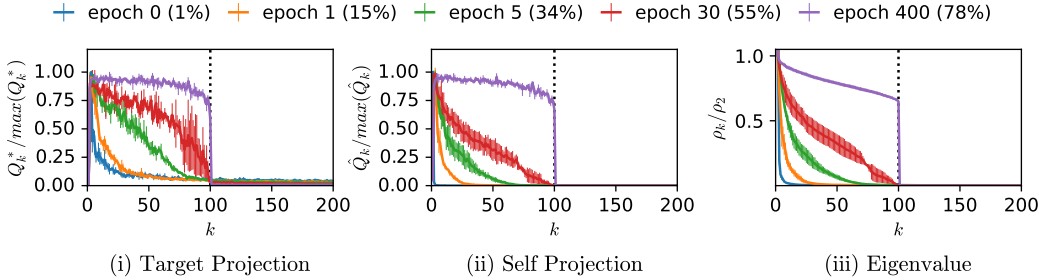

Figure 5: **Correlation among dynamics of feature quality, utilization, and intensity.** We show individual metrics (e.g., $Q^*(k) := \Pi^*(k) - \Pi^*(k-1)$) instead of cumulative metrics ($\Pi^*(k)$ and $\hat{\Pi}(k)$) at different epochs for ResNet18 on CIFAR-100, normalized for better presentation. Larger intensity features exhibit higher quality and utilization during training.

Figs. 3 and 4 illustrate how visualization complements the metric. In Fig. 3, the batch-norm included model uses only 100 features (eigenfunctions) in (ii) and shows clear low-rank structure in (iii). The batch normalization lacking model uses significantly more feature (4096 in total) in (ii) and show power-law distribution in (iii): indicating significantly lazy dynamics. In Fig. 4(ii), the model with the smallest learning rate uses over 100 features to express the learned function, unlike models with larger learning rates, suggesting lazier dynamics and poorer performance.

Fig. 5 reveals another novel pattern in feature learning dynamics: feature quality correlates with feature intensity during training, with larger features improving faster. While it is expected that a generalizing model in the rich regime obtains a few high-quality features after training, the correlation between quality and intensity *during* training has not been previously observed or studied. Interestingly, it differs from the silent alignment effect (Atanasov et al., 2021), where, in the dynamics of linear networks in a rich regime, quality precedes intensity.

## 6 DISCUSSION AND CONCLUSION

We introduced a performance-independent metric for dynamical richness, $\mathcal{D}_{LR}$, which reduces to neural collapse as a special case (Propositions 1 and 2) and empirically agrees with prior studies on lazy-to-rich transitions (Tables 2 and 3).

Table 3 and Fig. 2 provide examples of how the metric can be used for the analysis. We confirm established assumptions explicitly with an independent metric, while potentially challenging a few (Fig. 5). We also reveal new findings (Fig. 3), offering a direction for theoretical research.

Moreover, we provide a complementary visualization to extend these results for additional information, also uncovering novel insights such as the alignment between representation quality and feature intensity (Fig. 5).

**Interpretation of neural collapse.** We share our secondary findings on neural collapse. The functional utility of neural collapse remains an open question, particularly regarding its debated link to generalization (Hui et al., 2022; Kothapalli et al., 2022; Zhu et al., 2021). As established in Propositions 1 and 2, our metric is mathematically tied to neural collapse. Because our metric quantifies representational richness, this suggests neural collapse may serve fundamentally as an indicator of rich feature dynamics rather than generalization.

**Intermediate layers.** While our metric achieves computational efficiency by focusing exclusively on the last layer, this design inherently omits the dynamics of intermediate layers. In Section I, we share our exploration of the intermediate layers and the justification of our choice. Theoretically, our kernel-based framework assumes linearity between features and the learned function — a premise less applicable to intermediate layers subject to non-linear transformations — and empirically, intermediate layers yield negligible additional insight into dynamical richness. We attribute this to the last layer exhibiting the strongest low-rank bias, a view supported by the comparative rarity of deep neural collapse Rangamani et al. (2023) and the shallow nature of unlearning Jung et al. (2025). We leave the extension of this framework to deep feature learning dynamics for future work.

**Isotropic target function.** The current form of $\mathcal{T}_{MP}$ is limited to orthogonal and isotropic target functions. While this covers most classification tasks, a more general setup would be preferable. However, imbalanced label remains an active area of research in neural collapse (Fang et al., 2021; Yang et al., 2022; Hong & Ling, 2023; Yan et al., 2024; Gao et al., 2024), and we leave the extension for future work. See Section D.4 for further discussion.

**Conclusion.** By offering a lightweight, robust, and performance-independent metric on rich dynamics, we aim to lay the groundwork for future theoretical studies on their connection to representation learning. More broadly, we see this work as a diagnostic tool toward bridging empirical observations on representation learning (e.g., rule of thumb) and theoretical understanding of rich dynamics. In future work, we plan to extend beyond the current focus on balanced tasks.

## REPRODUCIBILITY STATEMENT

As mentioned in the main text, full experimental details, link to the source code, and statistical significance for the tables are provided in Section H. All error bars represent one standard deviation. The algorithm for approximating the metric and visualization is provided in Section E.

## ACKNOWLEDGMENT

We thank Charles London, Zohar Ringel, and Shuofeng Zhang for their helpful comments. NF acknowledges the UKRI support through the Horizon Europe guarantee Marie Skłodowska-Curie grant (EP/X036820/1). SL was supported by the National Research Foundation of Korea (NRF) grants No.2020R1A5A1016126 and No.RS–2024-00462910. CM acknowledges funding from an EPSRC iCASE grant with IBM (grant number EP/S513842/1). SH gratefully acknowledges partial support from NSF grants DMS-2209975, 2015341, 20241842, NSF grant 2023505 on Collaborative Research: Foundations of Data Science Institute (FODSI), the NSF and the Simons Foundation for the Collaboration on the Theoretical Foundations of Deep Learning through awards DMS-2031883 and 814639, and NSF grant MC2378 to the Institute for Artificial CyberThreat Intelligence and OperatioN (ACTION). The authors would also acknowledge support from His Majesty's Government in the development of this research.

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

# A   GLOSSARY

| Symbol. | Name | Definition | Ref |
|---|---|---|---|
| $C$ | Class count | The number of classes. | Section 2 |
| $p$ | Layer width | The width of the last layer | Section 2 |
| $n$ | Sample count | The number of training samples | Section 2 |
| $\mathcal{X}$ | Input space | Space of inputs | Section 2 |
| $q$ | Input distribution | The probability distribution that generates samples in the input space. | Section 2 |
| $\Phi$ | Feature map | A map from input to post-activation of the penultimate layer (the activations fed to the last layer). | Section 2 |
| $\Phi(x)$ | (Last layer) Features | A $p$-dimensional random variable or the post-activation of penultimate layer for $x \sim q$. | Section 2 |
| $\mathcal{T}$ | (Feature kernel) integral operator | A map from Hilbert space to Hilbert space that depends on $\Phi(x)$, $x \sim q$ (typically the last layer features) | Eq. (1) |
| $\mathcal{T}_{MP}$ | Minimum projection operator | A special set of operators that depends on the learned function $\hat{f}$. | Definition 1 |
| $e_k$ | $k^{\text{th}}$ eigenfunction | The eigenfunction of $\mathcal{T}$. | Eq. (2) |
| $\rho_k$ | $k^{\text{th}}$ eigenvalue | The eigenvalue of $\mathcal{T}$. | Eq. (2) |
| $\hat{f}$ | Learned function | A $C$-dimensional vector output function $\hat{f} : \mathcal{X} \to \mathbb{R}^C$ expressed by the neural network. | Section 2 |
| $f^*$ | Target function | A $C$-dimensional vector output function $f^* : \mathcal{X} \to \mathbb{R}^C$ with correct labels. The output is always a one-hot vector (up to a scaling constant). | Section 2 |
| $\mathcal{H}^*$ | Target function space | Space linearly spanned by the entries of the target function. | Section 3 |
| $\hat{\mathcal{H}}$ | Learned function space | Space linearly spanned by the entries of the learned function | Section 3 |
| $P_{\mathcal{H}}$ | Projection operator | Projection operator onto $\mathcal{H}$. If $\mathcal{H} = span\{e_1, e_2, \ldots, e_p\}$, where $e_k$'s are orthonormal, the projection operator is given as $P_{\mathcal{H}} = \sum_{k=1}^{p} |e_k\rangle \langle e_k|$. | Section 3 |
| $\alpha$ | Downscale constant | Target downscaling prefactor: $y \to y/\alpha$ | Section 4.1 |
| $\Sigma_b$ | Inter-class covariance matrix | Covariance of class mean vectors for the training set. | Eq. (34) |
| $\Sigma_W$ | Intra-class covariance matrix | Covariance of training feature vectors within given class. | Eq. (33) |
| $CKA(\cdot, \cdot)$ | Centered kernel alignment | Alignment measure between two matrices or operators. It is normalized between $[0, 1]$ and ignores the mean (for matrices) or constant function (for operators) | Eq. (7) |
| $\|\theta\|_F^2$ | Parameter norm | Norm of all parameters in the model. It was used as richness measure with smaller value meaning richer dynamics. | Section 4.1 |
| $\mathcal{D}_{LR}$ | low rank metric | Our proposed metric of dynamical richness. Smaller is richer. | Eq. (4) |
| $\mathcal{S}_{init}$ | Kernel deviation | The CKA measure between the learned and initial kernel. | Section 4.1 |
| $NC1$ | Neural collapse metric | The trace of $\Sigma_b^\dagger \Sigma_W$, measuring the training feature vectors variance compared to class boundaries. | Proposition 1 |
| $\Pi^*(k)$ | Cumulative quality | Measure of how well the first $k$ eigenfunctions span the target function space. | Eq. (5) |
| $\hat{\Pi}(k)$ | Cumulative utilization | Measure of how well the first $k$ eigenfunctions span the learned function space. | Eq. (5) |
| $Q^*(k)$ | Quality | Per-feature quality or $\Pi^*(k) - \Pi^*(k-1)$ where $\Pi^*(0) = 0$ | Fig. 5 |
| $\hat{Q}(k)$ | Utilization | Per-feature utilization or $\hat{\Pi}(k) - \hat{\Pi}(k-1)$ where $\hat{\Pi}(0) = 0$ | Fig. 5 |

## B  TECHNICAL SUPPLEMENTARY MATERIAL

Here, we introduce technical terms used in the main text and the following appendices.

### B.1  BRA-KET NOTATION

In physics, bra-ket notations are widely used to express the inner product in function space. In our paper, we use them to avoid overload of expectations and to clarify that we are using functions. Note that we use bra-ket notation for the expectation over the true input distribution only:

$$\langle f|g \rangle = \mathbf{E}_{x \sim q}[f(x)g(x)].$$

The notation is also useful for expressing operators such as $\mathcal{T}$:

$$\mathcal{T}[f] = \sum_{k=1}^{p} |\Phi_k\rangle \langle \Phi_k|f\rangle = [\Phi_1 \langle \Phi_1|f\rangle, \dots, \Phi_p \langle \Phi_p|f\rangle].$$

Because $\mathcal{T}$ maps vector function $f : \mathcal{X} \to \mathbb{R}^p$ to a vector function, $\mathcal{T}[f] : \mathcal{X} \to \mathbb{R}^p$ is also a vector function with

$$\mathcal{T}[f](x) = [\Phi_1(x) \langle \Phi_1|f\rangle, \dots, \Phi_p(x) \langle \Phi_p|f\rangle],$$

where $\langle \Phi_k|f\rangle$ are scalars and $\Phi_k : \mathcal{X} \to \mathbb{R}$ are functions.

### B.2  FEATURES ARE RANDOM VARIABLES AND (WELL-BEHAVED) RANDOM VARIABLES FORM A HILBERT SPACE

In machine learning, textbooks often overlook the mathematical distinction between feature maps and features, treating them as interchangeable. However, they are fundamentally different.

A feature map is a function $f : \mathcal{X} \to \mathcal{Y}$, defined independently of any distribution. A feature, by contrast, is a random variable induced by applying a feature map to inputs drawn from a distribution $q$ over $\mathcal{X}$.

Features are therefore distribution-dependent: applying the same feature map to different input distributions yields different features. For instance, a fixed neural network defines a feature map, but the resulting features — such as last-layer activations — will differ between MNIST and Fashion-MNIST due to changes in the input distribution. The distinction also applies to kernel operators $\mathcal{T} : \mathcal{H} \to \mathcal{H}$ (distribution dependent) and kernels $K : \mathcal{X} \times \mathcal{X} \to \mathbb{R}$ (distribution independent).

The distinction becomes more important when we wish to discuss Hilbert space. A Hilbert space requires an inner product between functions (feature maps), which depends on the underlying distribution. For example, $\sin(x)$ and $\sin(2x)$ are orthogonal for $q = unif[0, 2\pi]$ where $\int_0^{2\pi} \sin(x) \sin(2x) dx = 0$, but not for unit Gaussian distribution $q = \mathcal{N}(0, 1)$ where $\int_{-\infty}^{\infty} \sin(x) \sin(2x) e^{-x^2/2} dx \neq 0$.

For an underlying probability distribution and a set of functions (feature maps), we can form a Hilbert space. A common example of a Hilbert space is the set of solutions expressed by a linear model such as

$$f(x) = w_1 x + w_2 x^2 + w_3 x^3.$$

Assuming the features $[x, x^2, x^3]$ are linearly independent on input distribution $q$, they form a 3-dimensional Hilbert space.

### B.3  CENTERED KERNEL ALIGNMENT

The centered kernel alignment ($CKA$) (Kornblith et al., 2019) measures the similarity between two matrices or operators, and are commonly used for tracking the evolution of NTK (Baratin et al., 2021; Lou et al., 2022):

$$CKA(A, B) = \frac{\text{Tr}(c(A)c(B))}{\|c(A)\|_F \|c(B)\|_F}, \tag{7}$$

where $c(A) = (I - |\mathbf{1}\rangle \langle \mathbf{1}|)A(I - |\mathbf{1}\rangle \langle \mathbf{1}|)$ is a centering operator, where $I$ is identity, $|\mathbf{1}\rangle$ is the constant function or constant vector, and $\| \cdot \|_F$ is Frobenius norm. The centering operator removes the constant shift. The analogy is measuring the similarity of two multivariate Gaussian random variables by comparing their covariance matrices $\mathbf{E}[(X - \mathbf{E}[X])(X - \mathbf{E}[X])^T]$ instead of their autocorrelation matrices $\mathbf{E}[XX^T]$.

For $\mathcal{T}_{MP}$, if the learned function space $\hat{\mathcal{H}}$ contains the constant function — which is empirically true throughout the training for all our experiments and always true if it perfectly fits the training samples (Section D) — $c(\mathcal{T}_{MP})/\|c(\mathcal{T}_{MP})\|_F$ becomes an orthogonal projection operator on the constant complement of $\hat{\mathcal{H}}$, and is independent of the values of $a_1$ and $a_2$.

## B.4 EFFECTIVE DIMENSION

A covariance matrix or an integral operator may be fully ranked, but their eigenvalues may decay fast (e.g., by a power-law). The small eigendirections are often negligible, and researchers use various effective dimensions to measure the number of **significant** dimensions. In this appendix, we will use the exponent of entropy (Hill, 1973) to measure the effective dimension.

For a matrix or an operator with positive eigenvalues $[\rho_1, \cdots, \rho_p]$, the effective dimension is

$$\exp\left(-\sum_{i=1}^{p} \frac{\rho_i}{\sum_{j=1}^{p} \rho_j} \ln\left(\frac{\rho_i}{\sum_{j=1}^{p} \rho_j}\right)\right). \tag{8}$$

Note that the effective dimension, the exponential of Shannon entropy, is $d$ when the eigenvalues have $d$ equal non-zero entries. A slower decay of entries of $\rho$ (in non-increasing order of entries) generally yields a higher effective dimension than a vector with a faster decay.

The effective dimension of a matrix (operator) can be interpreted as the number of linearly independent vectors (functions) needed to effectively describe the matrix (operator). This is similar in spirit to the principal component analysis (PCA) (Abdi & Williams, 2010) in that only the directions with significant variance are considered.

As discussed in Section D, the first eigenfunction is always a constant function, which is irrelevant for our measure $\mathcal{D}_{LR}$ and dynamical interpretation. Because the first eigenvalue is often much larger than other significant eigenvalues, we use

$$D_{eff}(\rho) := 1 + \exp\left(-\sum_{i=2}^{p} \frac{\rho_i}{\sum_{j=2}^{p} \rho_j} \ln\left(\frac{\rho_i}{\sum_{j=2}^{p} \rho_j}\right)\right). \tag{9}$$

## C  LINEAR MODELS, FEATURES, AND THEIR INDUCTIVE BIAS

Here, we will introduce the recent findings on the inductive bias of linear models and demonstrate the significance of eigenvalues and eigenfunctions of $\mathcal{T}$. Linear regression or kernel regression is a rare case where its inductive bias is analytically calculable (Bordelon et al., 2020; Jacot et al., 2020; Spigler et al., 2020; Canatar et al., 2021; Cui et al., 2021; El Harzli et al., 2024; Simon et al., 2023). For a given feature map $\Phi : \mathcal{X} \to \mathbb{R}^p$ or kernel $K(x, x') = \Phi(x)^T \Phi(x')$, the model is expressed as the $p$ dimensional ($p$ can be infitnite) linear model

$$\hat{f}(x; w) = \sum_{k=1}^{p} w_k \Phi_k(x). \tag{10}$$

The objective of (ridgeless) regression is to minimize the following empirical loss

$$\mathcal{L}_{emp}(\hat{f}) = \frac{1}{2} \sum_{i=1}^{n} \left| \hat{f}(x^{(i)}) - f^*(x^{(i)}) \right|^2 + \lim_{\lambda \to 0} \lambda \|w\|_F^2, \tag{11}$$

where $x^{(i)}$ denotes the $i^{\text{th}}$ sample in the training set. The second term is the regularization which we take the limit so the solution is unique (pseudo inverse solution) for an overparameterized setup ($n < p$).

### C.1  EIGENFUNCTIONS - ORTHONORMALIZED FEATURES

In many cases, $[\Phi_1(x), \Phi_2(x), \cdots, \Phi_p(x)]$ are not orthonormal. For example, the features $[x, x^2]$ of $f(x) = w_1 x + w_2 x^2$ for $q = \text{Uniform}[0, 1]$ are not orthonormal:

$$\langle x | x^2 \rangle = \int_0^1 x^3 dx \neq 0, \quad \langle x | x \rangle = \int_0^1 x^2 dx \neq 1, \quad \langle x^2 | x^2 \rangle = \int_0^1 x^4 dx \neq 1. \tag{12}$$

The diagonalization into the eigenfunctions returns the orthonormal basis of the hypothesis (Hilbert) space $\mathcal{H} = \text{span}\{\Phi_1(x), \Phi_2(x), \cdots, \Phi_p(x)\}$, more formally known as Reproducing Kernel Hilbert Space (RKHS). The eigenvalue now shows the norm of the feature along the direction of eigenfunction as the features $[\Phi_1(x), \Phi_2(x), \cdots, \Phi_p(x)]$ are not normalized (Fig. 6).

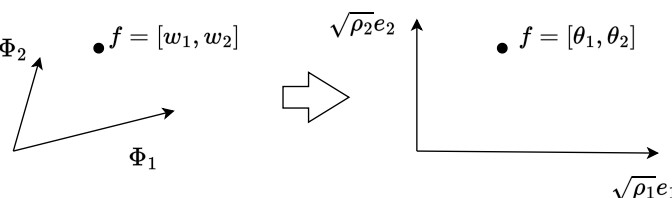

Figure 6: **Diagonalization of features** Since the features $[\Phi_1(x), \Phi_2(x), \cdots, \Phi_p(x)]$ span a vector space, each function $\Phi_k : \mathcal{X} \to \mathbb{R}$ can be represented as a vector. The overlap among $\Phi_k$'s (non-zero linear correlation) and differing norms of $\Phi_k$'s result in diagonalized features $[\sqrt{\rho_1}e_1, \sqrt{\rho_2}e_2, \cdots, \sqrt{\rho_p}e_p]$ to have distinct intensities (norms) $\rho_1, \cdots, \rho_p$.

The transformation between the eigenfunction basis and the feature basis is

$$\Phi_i(x) = \sum_j O_{ij} \sqrt{\rho_j} e_j(x), \qquad O_{ij} = \frac{1}{\sqrt{\rho_j}} \langle \Phi_i | e_j \rangle, \tag{13}$$

where $O \in \mathbb{R}^{p \times p}$ is an orthogonal matrix (follows trivially from Eq. (2)). Using Eq. (13), the model can be reparameterized in the eigenfunction basis

$$f(x) = \sum_{k=1}^{p} w_k \Phi_k(x) = \sum_{k=1}^{p} \theta_k \sqrt{\rho_k} e_k(x). \tag{14}$$

Note that because $\theta = Ow$, the norm of the parameters is conserved (i.e. $\|\theta\|_F = \|w\|_F$), indicating that weighted basis $[\sqrt{\rho_1}e_1, \sqrt{\rho_2}e_2, \cdots, \sqrt{\rho_p}e_p]$ instead of normalized basis $[e_1, e_2, \cdots, e_p]$ must be used to represent the 'intensity' or norm of the features $[\Phi_1(x), \cdots, \Phi_p(x)]$.

## C.2  INDUCTIVE BIAS TOWARD LARGE FEATURES FOR MINIMUM NORM SOLUTION

For overparameterized linear models, the inductive bias determines the returned function as many functions can express the training set. Here, we brief the linear models' inductive bias toward larger (significant) features or eigenfunctions (Bordelon et al., 2020; Jacot et al., 2020; Spigler et al., 2020; Canatar et al., 2021; Cui et al., 2021; El Harzli et al., 2024; Simon et al., 2023).

The inductive bias is quantified through learnability $L_k$ (Simon et al., 2023) or the expectation ratio between the learned coefficient and true coefficient for $e_k$:

$$L_k := \mathbf{E}_{S \sim q^n}\left[\frac{\langle \hat{f}|e_k\rangle}{\langle f^*|e_k\rangle}\right] = \frac{\rho_k}{\rho_k + \kappa}, \qquad \text{where} \quad \sum_{k=1}^{p} \frac{\rho_k}{\rho_k + \kappa} = n. \tag{15}$$

The learnabilities always sum to $n$ — the number of training datapoints — and constant $\kappa$ is the constant that satisfies the equality.

We can understand the inductive bias from the (vanishing) regularization on $L^2$ norm ($\lambda\|w\|_F^2$ in Eq. (11)) and parameterization in Eq. (14). If different eigenfunctions can equally express the training set, eigenfunction with larger eigenvalue often requires smaller coefficient $\theta_k$ to express the samples. Because $\|w\|_F^2 = \|\theta\|_F^2$, expressing the data with larger (eigenvalue) eigenfunctions minimizes the norm, creating an inductive bias toward larger features.

Instead of the derivation found in the references (Bordelon et al., 2020; Jacot et al., 2020; Spigler et al., 2020; Canatar et al., 2021; Cui et al., 2021; El Harzli et al., 2024) — which requires random matrix theory or replica trick — we show an example of the inductive bias in Fig. 7.

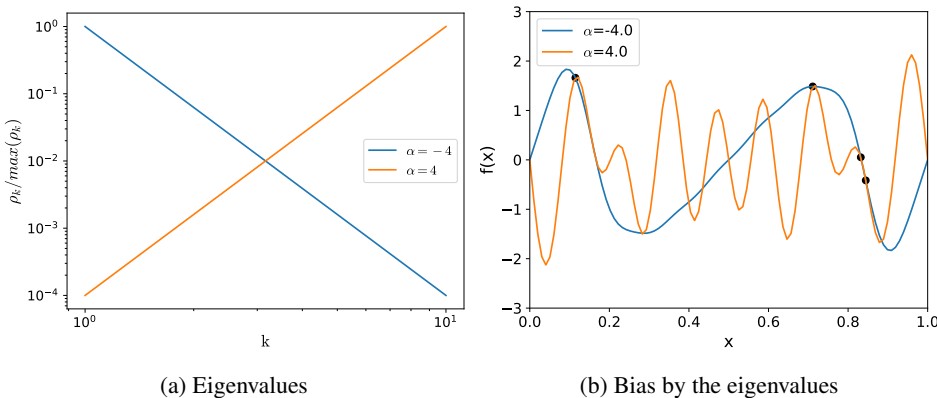

(a) Eigenvalues  (b) Bias by the eigenvalues

Figure 7: **Different inductive biases by the eigenvalues (intensity).** Two 10-parameter linear models are trained on 4 datapoints (overparameterized) with gradient flow. Both linear models use sin basis functions such that $f(x) = \sum_k^{10} w_k k^\alpha \sqrt{2} \sin(2\pi k x)$ — spanning an identical function space — but differ in the eigenvalues with $\alpha = -4$ (blue) and $\alpha = 4$ (orange), leading to different inductive biases. **(a):** The blue model has a greater intensity for lower-frequency sin functions, while the orange model shows the opposite. **(b):** The learned functions show that the blue model used lower-frequency functions while the orange used higher-frequency functions to express the training samples.

Fig. 7 shows that two linear models with same hypothesis space but different eigenvalues learn dramatically different functions. The blue model has large eigenvalues for low-frequency functions and fits the datapoints using low-frequency functions. The orange model with opposite eigenvalue distribution fits the training set with high-frequency functions — a clear example of an inductive bias toward large (intensity) features.

In the references, the generalization loss is

$$\mathcal{L}_G = \frac{1}{1 - \frac{1}{n}\sum_j^p L_j^2}\left(\sum_k^p (1 - L_k)^2 \langle e_k|f^*\rangle^{*2}\right). \tag{16}$$

The equation formalizes the task-model alignment: the alignment between large features (large $\rho_k$ thus large $L_k$) and high quality (large $\langle e_k|f^*\rangle$) leads to better generalization.

### C.3 DYNAMICAL INDUCTIVE BIAS TOWARD LARGE FEATURES

Here, we review how gradient descent introduces the dynamical inductive bias toward larger features. For a linear model with feature map $\Phi$ and kernel integral operator $\mathcal{T}$, the gradient flow dynamics under MSE loss with infinite training datapoints and zero initialization becomes

$$\frac{df}{dt} = -\mathcal{T}\left[f - f^*\right]. \tag{17}$$

The derivation is trivial with

$$\frac{df}{dt}(x') = \sum_k^p \frac{dw_k}{dt}\Phi_k(x') = \sum_k^p -\frac{\partial \mathcal{L}}{\partial w_k}\Phi_k(x') \tag{18}$$

$$= -\mathbf{E}_x\left[(f(x) - f^*(x))\sum_k^p \frac{df}{dw_k}(x)\right]\Phi_k(x') \tag{19}$$

$$= -\mathbf{E}_x\left[(f(x) - f^*(x))\sum_k^p \Phi_k(x)\right]\Phi_k(x') \tag{20}$$

$$= -\mathcal{T}\left[f - f^*\right], \tag{21}$$

where in the second line, we used Eq. (11) on infinite datapoints so $\mathcal{L}_{emp} = \mathcal{L}$. In the last line, we used the definition of integral operator (Eq. (1)).

Using the eigenvalues $\rho_k$ and eigenfunctions $e_k$ of the kernel integral operator $\mathcal{T}$, we obtain the dynamics for each $\langle e_k | f \rangle$ by taking inner products with $e_k$ on both sides of Eq. (17),

$$\frac{d\langle e_k | f \rangle}{dt} = \mathbf{E}_{x'}\left[e_k(x')\frac{df}{dt}(x')\right] = -\mathbf{E}_{x'}\left[e_k(x')\mathcal{T}\left[f - f^*\right](x')\right] \tag{22}$$

$$= -\langle e_k | \mathcal{T}[f - f^*] \rangle \tag{23}$$

$$= -\rho_k(\langle e_k | f \rangle - \langle e_k | f^* \rangle), \tag{24}$$

where we use the definition of eigenfunction in the last line (Eq. (2)). Expanding $f$ in the eigenfunction basis and plugging in Eq. (22), the function $f$ at time $t$ is a sum of $p$ independent modes that saturate faster for larger eigenvalues:

$$f(x)|_t = \sum_{k=1}^p \langle e_k | f^* \rangle \left(1 - e^{-\rho_k t}\right)e_k(x), \tag{25}$$

where we assumed $f$ is a zero function at initialization. Eq. (25) shows that gradient flow decouples the dynamics of linear models into $p$ modes, where each mode corresponds to the evolution of $\langle e_k | f \rangle$ having a saturation speed governed by $\rho_k$ — thus the dynamical inductive bias toward larger eigenfunction.

### C.4 APPLICATION TO OUR METHOD

Here, we detail the intuition of our visualization methods. As described in Bengio et al. (2013), feature learning (of representation) is providing **better** features for a simpler learner. The analogy allows us to describe a neural network as (1) feature map $\Phi$ providing better features for (2) the last layer — the simple learner (Fig. 8). Yet, **better** features for the last layer is ill-defined.

The last layer interacts with rest of the network only through the features. While the exact dynamics differs from linear models, the last layer maintains the dynamical inductive bias (Section C.3) toward larger features. Furthermore, the quality ($\langle e_k | f^* \rangle$), utilization ($\langle e_k | \hat{f} \rangle$), and intensity $\rho_k$ formalisms readily extend to the last layer features, motivating our visualization method.

In the rich regime, features evolve during training, and Fig. 9(a-c) demonstrates an example of change in features for 4-layer MLP trained to fit the Heaviside step function. The eigenfunctions are difficult to visualize in practice as the input space is high dimensional. Fig. 9(d-f) shows our visualization, capturing the evolution of features by measuring properties natural to linear regression.

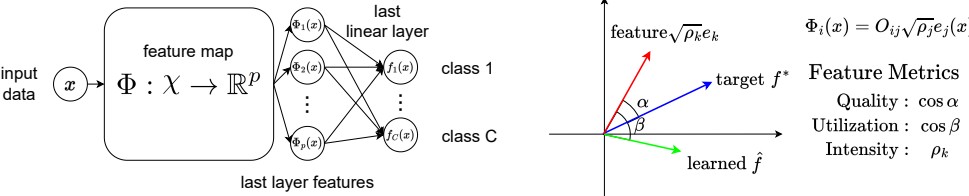

Figure 8: **A neural network decomposed into a feature map and a linear last layer. (left):**An abstract diagram depicting a DNN architecture as a combination of the feature map $\Phi : \mathcal{X} \to \mathbb{R}^p$ from the input space $\mathcal{X}$ to the $p$ post activations of the penultimate layer, and final linear classifier for $C$-way classification. Most neural networks share this abstract structure and mainly vary in how they create their feature maps. **(right):** Illustration of the visualization methods in function space.

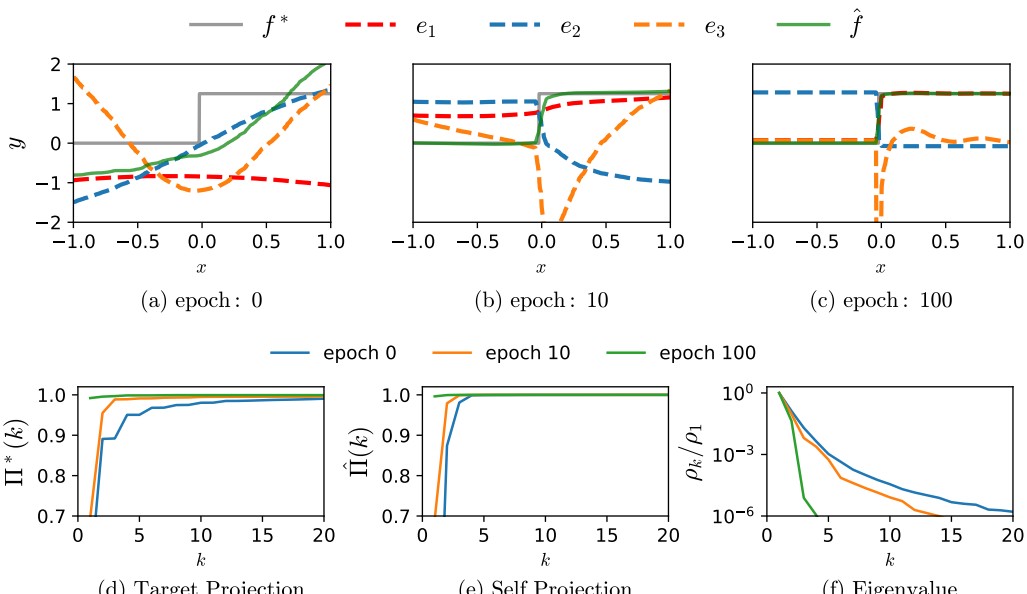

Figure 9: **Toy model demonstrating feature learning.** A 4-layer MLP with width 1000 and scalar input and output is trained to learn the Heaviside step function $f^*$ over the domain $[-1, 1]$. **(a):** The first three eigenfunctions (dashed) are shown at initialization, resembling orthogonal polynomials. **(b,c):** During training, features evolve such that a single feature (red) fits both the target and learned functions (grey and green). **(d,e,f):** Our visualization metrics defined in Eq. (5) at different epochs. (d) and (e) show that fewer features can express the target/learned function, and (f) shows that only a few features are significant after training.

# D THE RELATIONSHIP BETWEEN THE MP-OPERATOR $\mathcal{T}_{\mathbf{MP}}$ AND NEURAL COLLAPSE

Neural collapse (Papyan et al., 2020) (NC) refers to a state of a DNN when the feature vectors of the training set and last layer weights form a symmetric and clustered structure at the terminal phase of training (TPT) or when trained past the point where the training error vanishes. The structure has been studied mainly in the Unconstrained Feature Model (UFM) (Mixon et al., 2020) and has sparked theoretical studies. See Kothapalli et al. (2022) for a review.

NC is defined by the emergence of four interconnected phenomena upon TPT: NC1) collapse of within-class variability, NC2) convergence of features to a rigid simplex equiangular tight frame (ETF) structure, NC3) alignment of the last layer and features, and NC4) simplified decision by nearest class:

1. **Within class variance tends to 0**

$$\Sigma_b^\dagger \Sigma_W \to 0, \tag{26}$$

2. **Convergence to simplex ETF**

$$\frac{(\mu_i - \bar{\mu})^T (\mu_j - \bar{\mu})}{\|(\mu_i - \bar{\mu})\|_2 \|(\mu_j - \bar{\mu})\|_2} \to \frac{C\delta_{ij} - 1}{C - 1}, \tag{27}$$

3. **Convergence to self duality**

$$\frac{w_i}{\|w_i\|_2} - \frac{\mu_i - \bar{\mu}}{\|(\mu_i - \bar{\mu})\|_2} \to 0, \tag{28}$$

4. **Simplification to nearest class center**

$$\arg\max_i w_i \Phi(x) + b_i \to \arg\min_i \|\Phi(x) - \mu_i\|_2. \tag{29}$$

## D.1 PROOFS OF PROPOSITION 1 AND 2

For completeness, we restate the conditions. We assume the true distribution $q$ equals the empirical distribution of the training set — the expectation $\mathbf{E}_x$ (and bra-ket notation) is over the training samples. We assume balanced classification and define $A_i$ as the set of training samples with class label $i$. We assume the learned function is a perfectly classifying indicator function, giving

$$\hat{f}_i(x) := \begin{cases} 0 & : x \notin A_i \\ 1 & : x \in A_i \end{cases} \tag{30}$$

The feature class mean vector for label $i$ is

$$\mu_i = \mathbf{E}_{x \in A_i}[\Phi(x)] = \frac{C}{n} \sum_{x \in A_i} \Phi(x) = C \langle \hat{f}_i | \Phi \rangle. \tag{31}$$

Note that $\langle \hat{f}_i | \Phi \rangle \in \mathbb{R}^p$ is a vector with $\langle \hat{f}_i | \Phi \rangle = [\langle \hat{f}_i | \Phi_1 \rangle, \ldots, \langle \hat{f}_i | \Phi_p \rangle]$.

The global mean vector is

$$\bar{\mu} = \frac{1}{C} \sum_i^C \mu_i = \langle \Phi | \sum_{i=1}^C \hat{f}_i \rangle = \langle \mathbf{1} | \Phi \rangle. \tag{32}$$

The feature intra (within)-class covariance $\Sigma_W \in \mathbb{R}^{p \times p}$ is

$$\Sigma_W := \sum_i^C \mathbf{E}_{x \in A_i} \left[ (\Phi(x) - \mu_i)(\Phi(x) - \mu_i)^T \right]. \tag{33}$$

The feature inter (between)-class covariance $\Sigma_b \in \mathbb{R}^{p \times p}$ is

$$\Sigma_b := \frac{1}{C} \sum_i^C (\mu_i - \bar{\mu})(\mu_i - \bar{\mu})^T. \tag{34}$$

**Lemma 1.** *For the conditions given above, if $\mathcal{T}[f](x') = \Phi(x')^T \mathbf{E}_{x \sim q'}[\Phi(x)f(x)]$ is $\mathcal{T}_{MP}$, then*

$$\Phi(x')^T(\mu_i - \bar{\mu}) = a_2(C\hat{f}_i(x') - 1). \tag{35}$$

*Proof.* By the definition of indicator functions of equal partitions,

$$\langle \hat{f}_i | \hat{f}_j \rangle = \frac{1}{C}\delta_{ij}, \qquad \sum_{i=1}^{C} \hat{f}_i = \mathbf{1}, \qquad \langle \mathbf{1} | \hat{f}_j \rangle = \frac{1}{C}. \tag{36}$$

It follows that $C\hat{f}_j - \mathbf{1}$ is orthogonal to $\mathbf{1}$:

$$\langle \mathbf{1} | C\hat{f}_i - \mathbf{1} \rangle = 0. \tag{37}$$

We can express the function $\Phi(x')^T(\mu_i - \bar{\mu}) : \mathcal{X} \to \mathbb{R}$ in terms of $\mathcal{T}$

$$\Phi(x')^T(\mu_i - \bar{\mu}) = \Phi(x')^T \left( \langle \Phi | C\hat{f}_i \rangle - \langle \Phi | \mathbf{1} \rangle \right) \tag{38}$$

$$= \Phi(x')^T \mathbf{E}_x[\Phi(x)(C\hat{f}_i(x) - 1)] \tag{39}$$

$$= \mathcal{T}[C\hat{f}_i - 1](x'), \tag{40}$$

where we used the definition of the means (Eqs. (31) and (32)) in the first line and the definition of $\mathcal{T}$ (Eq. (1)) in the last line. As $\mathcal{T}$ is $\mathcal{T}_{MP}$, we obtain

$$\Phi(x')^T(\mu_i - \bar{\mu}) = \mathcal{T}_{MP}[C\hat{f}_i - 1](x') \tag{41}$$

$$= a_2(C\hat{f}_i(x') - 1), \tag{42}$$

where we used Eq. (37) and the definition of $\mathcal{T}_{MP}$ (Definition 1). $\qquad \square$

**Corollary 1.** *In the setup of Lemma 1, we have*

$$\mu_i^T(\mu_j - \bar{\mu}) = a_2(C\delta_{ij} - 1).$$

*Proof.* From the definition of class mean (Eq. (31)) and Lemma 1,

$$\mu_j^T(\mu_i - \bar{\mu}) = \mathbf{E}_{x'}\left[ C\hat{f}_j(x')\Phi(x')^T(\mu_i - \bar{\mu}) \right] \tag{43}$$

$$= \mathbf{E}_{x'}\left[ C\hat{f}_j(x')a_2(C\hat{f}_i(x') - 1) \right] \tag{44}$$

$$= a_2(C\delta_{ij} - 1), \tag{45}$$

where we used that $\mathbf{E}_{x'}[\hat{f}_j^2(x')] = \mathbf{E}_{x'}[\hat{f}_j(x')] = C^{-1}$. $\qquad \square$

**Proposition 1.** *If $\mathcal{T}$ is an MP-operator, then the **NC1** condition (collapse of within-class variability) $\Sigma_b^\dagger \Sigma_W = 0$ holds, where inter-class covariance matrix $\Sigma_b$ and intra-class covariance matrix $\Sigma_W$ are*

$$\Sigma_b = \sum_{j=1}^{C}(\mu_j - \bar{\mu})(\mu_j - \bar{\mu})^T \quad and \quad \Sigma_W = \sum_{i=1}^{C} \mathbf{E}_{x \in A_i}\left[ (\Phi(x) - \mu_i)(\Phi(x) - \mu_i)^T \right].$$

*Proof.* We will show that $\mathrm{Tr}(\Sigma_b \Sigma_W) = 0$, which automatically proves the proposition as covariance matrices are positive semi-definite. We have

$$\mathrm{Tr}(\Sigma_b \Sigma_W) = \mathrm{Tr}\left( \sum_j (\mu_j - \bar{\mu})(\mu_j - \bar{\mu})^T \sum_i \mathbf{E}_{x \in A_i}(\Phi(x) - \mu_i)(\Phi(x) - \mu_i)^T \right) \tag{46}$$

$$= \sum_{i,j} \mathbf{E}_{x \in A_i}\left[ \left( (\Phi(x) - \mu_i)^T(\mu_j - \bar{\mu}) \right)^2 \right] \tag{47}$$

$$= \sum_{i,j} \mathbf{E}_{x \in A_i}\left[ a_2^2 \left( C\hat{f}_j(x) - 1 - C\delta_{ij} + 1 \right)^2 \right] \tag{48}$$

$$= a_2^2 C^2 \sum_{i,j} \mathbf{E}_{x \in A_i}\left[ \left( \hat{f}_j(x) - \delta_{ij} \right)^2 \right], \tag{49}$$

where we used linearity of trace, sum, expectation in the first line and Lemma 1 in the second line. Expanding Eq. (49),

$$\text{Tr}(\Sigma_b \Sigma_W) = a_2^2 C^2 \sum_{i,j} \mathbf{E}_{x \in A_i} \left[ (1 - 2\delta_{ij}) \hat{f}_j(x) + \delta_{ij} \right] \tag{50}$$

$$= a_2^2 C \sum_{i,j} (\delta_{ij}(2 - 2\delta_{ij})) \tag{51}$$

$$= 0, \tag{52}$$

where we used that learned functions are indicator functions in the second line. □

**Proposition 2.** *If $\mathcal{T}$ is an MP-operator, then the **NC2** condition (convergence of features to a simplex equiangular tight frame) holds:*

$$(\mu_i - \bar{\mu})^T (\mu_j - \bar{\mu}) \propto \delta_{ij} - \frac{1}{C}.$$

*Proof.* From the definition of global mean (Eq. (32)) and Lemma 1,

$$\bar{\mu}^T (\mu_i - \bar{\mu}) = \mathbf{E}_{x'} \left[ \Phi(x')^T (\mu_i - \bar{\mu}) \right] \tag{53}$$

$$= \mathbf{E}_{x'} \left[ a_2 (C \hat{f}_i(x') - 1) \right] \tag{54}$$

$$= 0. \tag{55}$$

Using Corollary 1 and Eq. (55),

$$(\mu_i - \bar{\mu})^T (\mu_j - \bar{\mu}) = C a_2 (\delta_{ij} - \frac{1}{C}). \tag{56}$$

□

## D.2 The first eigenfunction, the constant function, the analog of global mean vector in simplex ETF

Here, we discuss the constant function, which, by definition of $\mathcal{T}_{MP}$ (Definition 1), is the largest eigenfunction if $\mathcal{T}$ is $\mathcal{T}_{MP}$. In all experiments except Fig. 9, the largest eigenfunction is the constant function throughout training, with $\langle \mathbf{1} | e_1 \rangle > 0.95$. For our visualization, we ignore the first eigenvalue from analysis as the constant function is removed in CKA (Eq. (7)).

In the NC2 condition, the $k$-simplex ETF structure is a set of $k$ orthogonal vectors projected on $k - 1$ dimensional space along the complement of the global mean vector. For example, projecting $[0, 0, 1]$, $[0, 1, 0]$, and $[1, 0, 0]$ onto the orthogonal complement of $[\frac{1}{3}, \frac{1}{3}, \frac{1}{3}]$ (the global mean vector) gives three vertices $[-\frac{1}{3}, -\frac{1}{3}, \frac{2}{3}]$, $[-\frac{1}{3}, \frac{2}{3}, -\frac{1}{3}]$, $[\frac{2}{3}, -\frac{1}{3}, -\frac{2}{3}]$ which forms an (ordinary plane) equilateral triangle.

Our operator $\mathcal{T}_{MP}$ (Definition 1) has an analogous structure, where it is a (scaled) projection operator ($a_2 P_{\hat{\mathcal{H}}}$) except along the constant function. The $CKA$ (Eq. (7)) also measures the alignment except along the mean direction (for matrices) or the constant direction (for operators). This allows neural networks to have any arbitrary eigenvalue for the constant function, yet perfectly align with $\mathcal{D}_{LR} = 0$.

When $\mathcal{T}$ is $\mathcal{T}_{MP}$, the definition of $\mathcal{T}_{MP}$ (Definition 1) trivially shows that the largest eigenfunction is the constant function. A peculiar observation is that the first eigenfunction of $\mathcal{T}$ remains the constant function ($\langle \mathbf{1} | e_1 \rangle > 0.95$) **throughout training** in all our experiments (except Fig. 9). This pattern also appears in hierarchical datasets (Saxe et al., 2022), suggesting that natural data may consistently prioritize the constant function as the dominant mode — warranting further investigation.

## D.3 Generality of our metric beyond neural collapse

As discussed in the main text, our metrics are defined in terms of functions (random variables) while neural collapse is defined by the feature vectors of the training set. Working in the function space, we can measure the quality Eq. (5), which is inaccessible from formalism using training feature vectors.

Another important difference is that we focus on the learned function instead of mean class label vectors. This allows more independence beyond neural collapse, allowing the method to be applied on broader set of tasks: tasks with orthogonal and isotropic target functions form a broader class than balanced classification tasks. Regression task with scalar output is an example.

In Fig. 10, we train ResNet18 on MNIST as a **regression** task with scalar output ($f^* : \mathcal{X} \to \mathbb{R}$), where a digit $i$ is correctly predicted if $i - 0.5 < f(x) < i + 0.5$. Unlike $C$-way classification, which typically uses $C$ features, the regression model reaches the rich regime with just two — one of which is constant. Neural collapse formalism, focusing on class labels, does not readily extend to regression tasks, demonstrating that our function-based formalism is more general, label-independent, and applicable beyond classification.

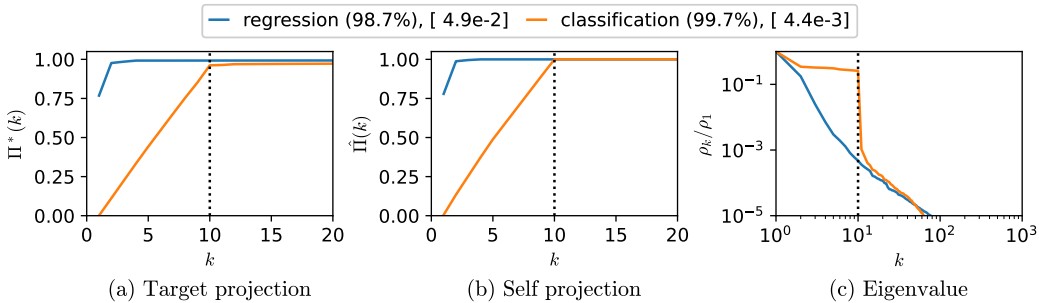

|  (a) Target projection | (b) Self projection | (c) Eigenvalue |

Figure 10: **Rich dynamics for regression problem with scalar output**. ResNet18 was trained on MNIST via regression (blue) and classification (orange). For regression, the target/learned functions are scalar output functions, and an image $x$ with digit $i$ is correctly classified if $i - 0.5 < f(x) < i + 0.5$. It can be seen that both models are in the rich regime, using the minimum number of features.

Finally, the independence from the class labels allows another benefit of being applicable during training. Neural collapse is defined only during the terminal stage of training and assumes 100% classification. A measure without perfect classification can lead to unstable values, typically rising from the pseudo-inverse of $\Sigma_b$ (See also Section H). The NC1 measure shows a dramatic change in logarithmic scale for experiments in Fig. 11(a,b) while our measure is more robust to such differences.

### D.4 ISOTROPICITY ASSUMPTION

As stated at the beginning of Section 2 and in the limitation paragraph, our paper's scope is on isotropic outputs. This is a slightly weaker assumption compared to prevailing standards in neural collapse literature (Papyan et al., 2020; Mixon et al., 2020; Han et al., 2022; Jacot et al., 2025), as discussed above. Because simplex ETF structures rely significantly on the symmetry, proposing a plausible extension for imbalanced tasks is an active area of research on its own (Fang et al., 2021; Yang et al., 2022; Yan et al., 2024; Gao et al., 2024). For example, imbalanced targets suffer from the minority collapse where multiple classes collapse to a single vector for a finite dataset (Fang et al., 2021). This becomes particularly a problem for a long-tailed target (e.g., natural language dataset), which is also an active area of research (Gao et al., 2024).

As another challenge, the eigenvalue distribution of features now depends on many factors such as weight decay. Assume a diagonal last layer with $p = C$. Assuming anisotropic target $f_k^*(x) = a_k e_k(x)$, the intensity $a_k$ can either come from the $\|\phi_k(x)\|$ or the last layer weight $w_k$. In other words,

$$a_k = w_k \|\phi_k(x)\|. \tag{57}$$

For strong weight decay, the weights $[w_1, \ldots, w_p]$ become more isotropic and anisotropy are absorbed in $\|\phi_k(x)\|$. For two-layer diagonal linear networks in rich regime with isotropic input ($x_k \sim \mathcal{N}(0, 1)$), $w_k \propto |\phi_k(x)|$ where anisotropy is distributed evenly between the weights and features. Since our metric compares $\phi(x)$ to $f(x)$, these differences, though yielding the same output, returns different levels of dynamical richness. But then, what representation shapes are favored by rich dynamics? While symmetry from an isotropic target and normalized CKA have let us bypass this issue, introducing anisotropy and factors such as weight decay complicates the dynamics and requires more concise understanding of the rich dynamics.

## E CALCULATING THE METRIC AND VISUALIZATION IN PRACTICE

### E.1 METRIC

In the main text, we used function and operator formalism. In practice, we must use finite approximation. We can use the CKA equation Eq. (7)

$$CKA(A, B) = \frac{\text{Tr}(c(A)c(B))}{\|c(A)\|_F \|c(B)\|_F},$$

(58)

but use $n \times n$ feature matrices for $A$ and $B$ instead of operators. Let $A^{1/2} \in \mathbb{R}^{p \times n}$ be feature vectors (of the penultimate layer) of $n$ samples and $B^{1/2} \in \mathbb{R}^{c \times n}$ be output vectors (of the network) for $n$ samples. We can then approximate CKA by inserting $A = (A^{1/2})^T A^{1/2}$ and $B = (B^{1/2})^T B^{1/2}$ into the above equation. Because we already have the square root matrices, the computation cost of CKA is $\mathcal{O}(npc)$.

Recall that $n$ can be any number, and can be sampled from both train and test set. As $A$ and $B$ are at most rank $p$ and $c$ respectively, $n > p$ suffices in practice. Because $p \approx 10^3$ in many models, we can expect $\mathcal{O}(p^2 c)$ computation. In an author's laptop, it can be computed in less than a minute for all experiments. For NTK, $p$ is no longer the width of the last layer, but total number of parameters, quickly becoming infeasible.

### E.2 VISUALIZATION

**Time complexity.** The algorithm is computationally efficient in practice. The main cost in computing Eq. (6) is the SVD, typically with time complexity $\mathcal{O}(p^3)$. A good approximation requires only $n = \mathcal{O}(p)$ datapoints—more than the last layer width. Performing SVD on $\phi$ instead of $\phi\phi^T$ yields $U$ and $\sqrt{S}$ with $\mathcal{O}(p^3)$ cost. Because last layers are typically narrow (e.g., $p \approx 10^3$) and the algorithm requires just $n = \mathcal{O}(p)$ forward passes, the overall computation is lightweight. In an author's laptop, the computation for all experiments takes less than a few minutes.

## F    EXAMPLES OF OUR VISUALIZATION METHOD

In this section, we report additional findings made with our visualization tools.

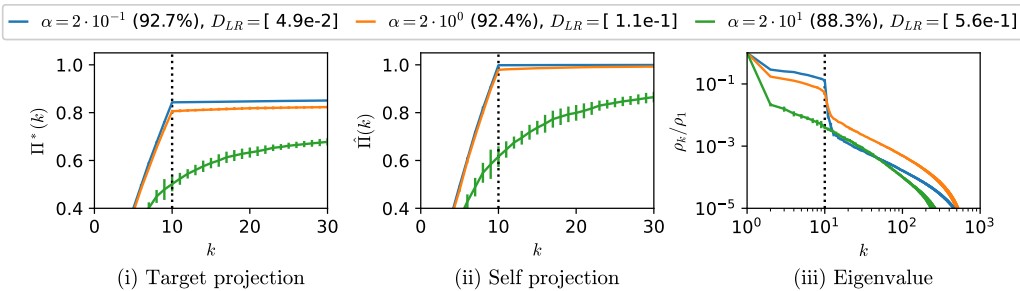

Figure 11: **Lazy/rich transition in target downscaling**, illustrating how lazier dynamics (ii) use more features and (iii) show a slower decay of eigenvalues (ignoring $\rho_1$ of constant function, see Section D).

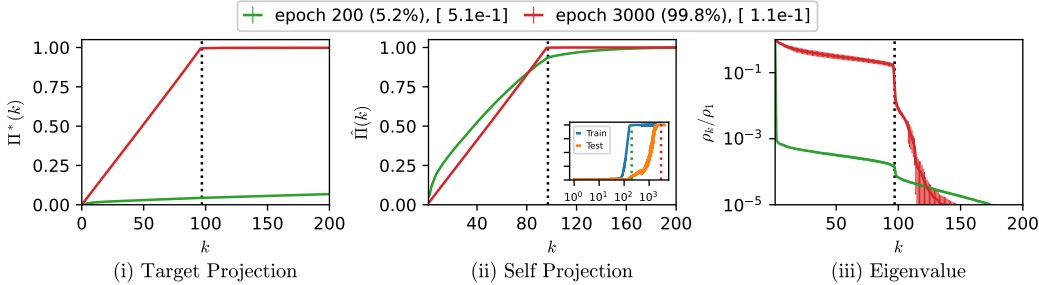

Figure 12: **Lazy/rich transition in grokking**. A 2-layer transformer is trained on the modular $p$ division task. The inset in the middle shows the training and test accuracies, where green and red vertical lines indicate before and after grokking (steps 200 and 3000). Our metric in square brackets shows the transition into the rich regime, while our visualization shows a clear difference in the use of features.

**Low rank bias.**    Table 4 shows that low-rank inductive bias is strong in vision tasks, where the number of significant features always equals $C$ (the number of orthogonal functions). See Eq. (9) in Section B.4 for the $D_{eff}(\rho)$ expression. We also observe that our measure $\mathcal{D}_{LR}$ is small in these setups.

Table 4: Metrics of ResNet18 and VGG16 trained on image datasets

| model | dataset | $D_{eff}(\rho)$ | $D_{eff}(\hat{Q})$ | $\mathcal{D}_{LR}$ | test accuracy (%) |
|---|---|---|---|---|---|
| ResNet18 | MNIST | 10.1 | 10.0 | $4.4 \cdot 10^{-3}$ | 99.7 |
| VGG16 | MNIST | 10.0 | 10.0 | $1.8 \cdot 10^{-2}$ | 99.4 |
| ResNet18 | CIFAR10 | 10.4 | 10.0 | $3.1 \cdot 10^{-3}$ | 94.8 |
| VGG16 | CIFAR10 | 10.0 | 10.0 | $1.1 \cdot 10^{-3}$ | 93.3 |
| ResNet18 | CIFAR100 | 99.9 | 99.7 | $2.5 \cdot 10^{-2}$ | 78.3 |
| VGG16 | CIFAR100 | 95.8 | 99.4 | $7.4 \cdot 10^{-2}$ | 71.9 |

**Weight decay is not the source of inductive bias.**    Fig. 13 shows that weight decay indeed aids low-rank representation and smaller $\mathcal{D}_{LR}$. However, the model without any weight decay already shows sufficiently low-rank representations, suggesting that dynamical inductive bias, not the weight decay is the main driving cause of low-rank representations.

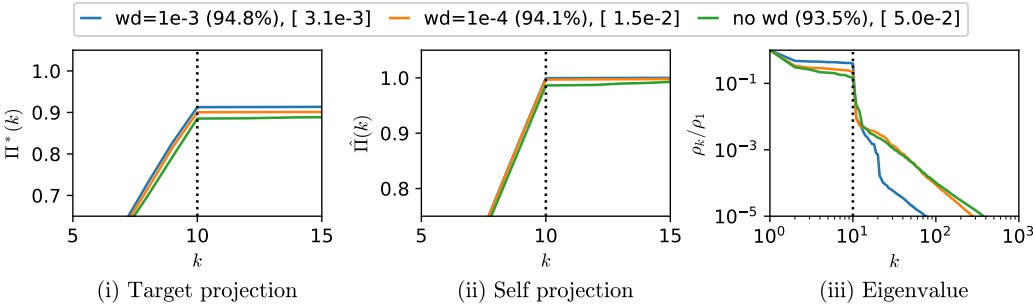

Figure 13: **Effect of weight decay on rich dynamics.** ResNet18 trained on CIFAR10 with varying weight-decay (shown relative value to the fixed learning rate of $0.05$). Larger weight decay leads to richer dynamics with smaller $\mathcal{D}_{LR}$, but the dynamics is already rich without the weight decay.

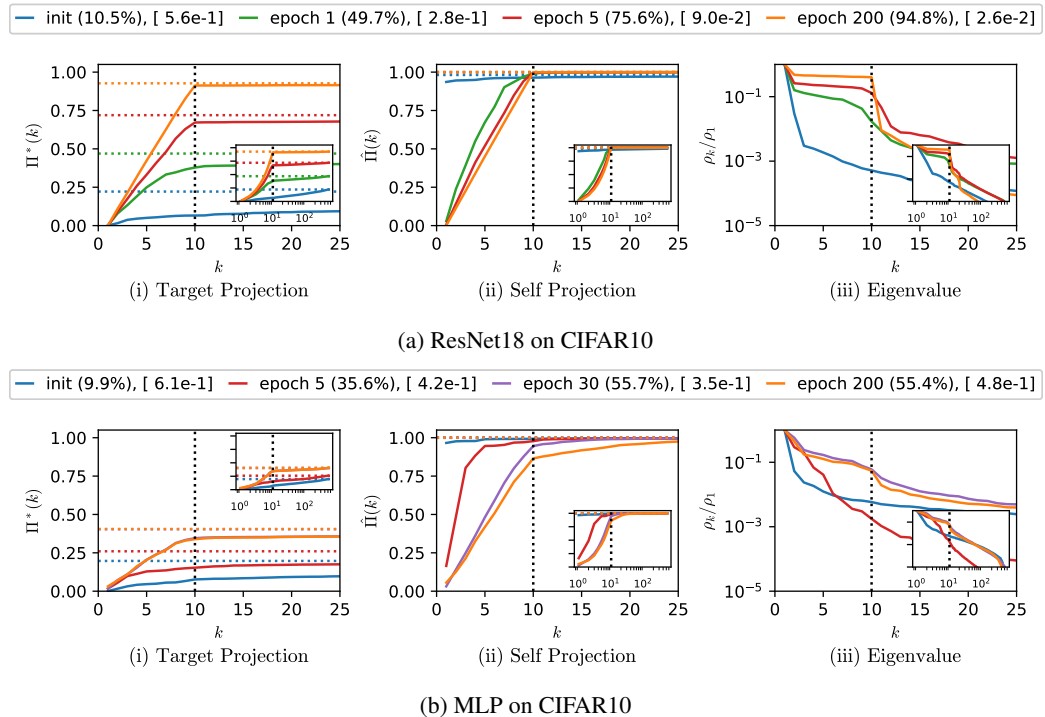

Figure 14: **Role of architecture in the richness of the dynamics.** ResNet18 (a) and a 4-layer MLP of width 512 (b) are trained on CIFAR10, and their metrics are shown at different epochs. ResNet18 concentrates the metrics on the first 10 features after just one epoch, which persists until the end of training. In contrast, MLP shows a less dramatic concentration on the first 10 features and deviates from the rich dynamics around epoch 30 when it begins to use more features.

**Role of architecture.** As studied in Saxe et al. (2022), the architecture and dataset pair can influence the performance even when dynamics are greedy (race toward shared representation). In Fig. 14, we examine the dynamics of ResNet18 and a 4-layer MLP trained on CIFAR-10. In Fig. 14(a), the training dynamics is restricted to the first 10 significant features, consistent with theoretical work in linear neural networks. In Fig. 14(b), the training is mainly focused on the first 10 features, but we observe that more features are used as training progresses, more similar to the dynamics of linear models. We speculate that the lack of ability to feature learn in the earlier layers leads to the use of additional features and thus leads to lazier training dynamics.

**Rich dynamics $\neq$ generalization.** In Fig. 1, we showed an example where rich dynamics led to poorer performance. Fig. 16 shows a similar experiment in which both rich and lazy models achieve near-perfect performance ($> 95\%$ test accuracy). Our visualization clearly shows the different usages of features even when the performances are similar.

**Rich dynamics without underlying data structure.** Fig. 15 illustrates that rich dynamics can occur independently of representation enhancement or underlying data structure. We trained ResNet18 on CIFAR-10 with varying levels of label shuffling and observed that even with fully randomized labels, the model enters the rich regime. This suggests that the dynamical low-rank bias is strong enough to collapse expressive networks into minimal representations, consistent with prior observations of neural collapse under random labeling (Zhu et al., 2021; Hui et al., 2022).

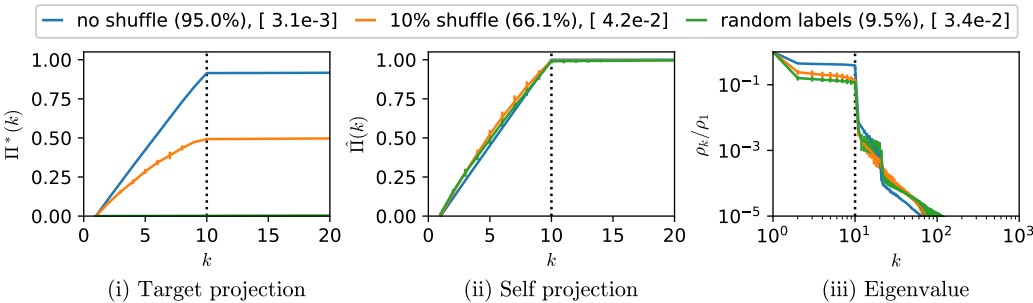

Figure 15: **ResNet18 on $10^4$ CIFAR-10 datapoints with shuffled labels.** Models are trained with 0% (blue), 10% (orange), and 100% (green) label shuffling. As shown in the square brackets, all exhibit rich dynamics with $\mathcal{D}_{LR} < 0.1$. Visualizations in (ii) and (iii) confirm the use of the top 10 significant features, but varying feature qualities (i) lead to varying test accuracies, shown in parentheses.

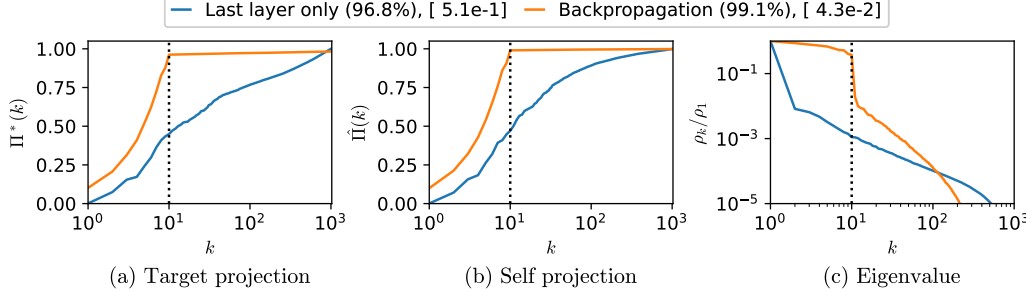

Figure 16: **Generalization does not imply rich regime.** A CNN with width $p = 1024$ is trained on the full MNIST dataset using two dynamics: last-layer-only training (lazy) and full backpropagation (rich). Although both dynamics achieve test accuracy above $95\%$, they lead to dramatically different use/significance of features: the metrics concentrate on the first 10 features for the rich regime, and they are more evenly spread for the lazy regime.

**Cross entropy loss.** Fig. 17 shows that our method extends to models trained on cross-entropy loss, while the mathematical interpretation is less straightforward.

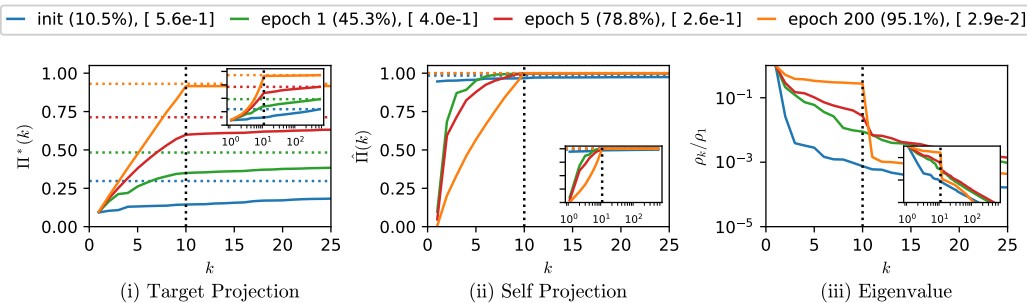

(i) Target Projection      (ii) Self Projection      (iii) Eigenvalue

Figure 17: **Training with cross-entropy loss.** We apply our visualization method to a ResNet18 trained on CIFAR-10 with cross-entropy loss. Similar to the MSE-trained model, it achieves small value of $\mathcal{D}_{LR}$ and shows concentration of feature usage and significance See Fig. 2(a) for comparison. The notable difference is in the speed difference among the first $C$ eigenfunctions. In MSE, the first $C$ eigenfunctions show nearly isotropic growth from epoch 1, while for CE, there is a notable difference in $\hat{\Pi}(k)$ and $\rho_k$ depending on $k$ until epoch 5. This may arise from the non-linearity coming from Cross entropy. Insets display all 512 features rather than just the top 25.

## G    RELATED WORKS ON VISUALIZATION

Eigendecomposition has long been a standard tool in science and engineering (Buckley & Xu, 1990; Amadei et al., 1993; te Vrugt & Wittkowski, 2020), demonstrating its value for visualization. Building on this, related empirical works have appeared (Raghu et al., 2017; Morcos et al., 2018; Kornblith et al., 2019). For example, Singular vector canonical correlation analysis (Raghu et al., 2017) resembles eigendecomposition but focuses on performance along leading directions rather than on the number of features used to represent the learned function (i.e., lacking $\hat{\Pi}$). (Morcos et al., 2018; Kornblith et al., 2019) compare different networks, whereas our focus is on before- and after-last-layer activations. Importantly, these approaches were not developed as visualization tools for analyzing rich dynamics.

Similarly, visualizations related to neural collapse use eigendecomposition (Papyan, 2020), but are inherently class-dependent, whereas our metric and visualization are class-independent. See Section D.3 and Fig. 10.

The closest line of work is kernel-based visualization (Bordelon et al., 2020; Canatar et al., 2021; Simon et al., 2023), especially (Canatar & Pehlevan, 2022). However, their visualization focuses on cumulative power and the eigenvalue spectrum, whereas ours additionally separates dynamics from performance, providing a complementary perspective. Additionally, the rich dynamics in the literature refers to change in NTK, not the collapse (low-rank bias) of feature kernel.

While visualization is part of our contribution, our main contribution lies in introducing $\mathcal{D}_{LR}$ as a metric of richness and presenting supporting empirical results. Visualization serves as a complementary tool to illustrate the dynamics and facilitate future studies.

# H    EMPIRICAL DETAILS

Here, we share information on our practical setups and their statistical significance. The code can be found at `https://github.com/yoonsoonam119/PrEF`.

## H.1    STATISTICAL SIGNIFICANCE

We reproduce tables from the main text but with the addition of one standard deviation values. In the main text, all error bars in the plots represent one standard deviation over at least three runs.

Table 5: Statistical significance of Figure 1(b)

|  | Full backprop (Rich) | Last layer only (Lazy) |
|---|---|---|
| Train acc. $\uparrow$ | 100 $(\pm 0)\%$ | 98.7 $(\pm 0.11)$ % |
| Test acc. $\uparrow$ | **10.0** $(\pm 4.3 \cdot 10^{-2})\%$ | **74.4** $(\pm 0.19)$ % |
| $\mathcal{D}_{LR} \downarrow$ | $8.7 \cdot 10^{-3}$ $(\pm 4.0 \cdot 10^{-4})$ | $6.3 \cdot 10^{-1}$ $(\pm 3.2 \cdot 10^{-3})$ |

Table 6: Statistical significance of Table 1

| Epoch | 0 (init) | 200 |
|---|---|---|
| Train Acc.$\uparrow$ | 10.0 $(\pm 0.8)\%$ | 10.2$(\pm 0.27)\%$ |
| Test Acc.$\uparrow$ | 9.85 $(\pm 1.0)\%$ | 10.0$(\pm 0.33)\%$ |
| $\mathcal{D}_{LR}\downarrow$ | **0.59** $(\pm 1.6 \cdot 10^{-2})$ | **1.0** $(\pm 6.4 \cdot 10^{-5})$ |
| $\mathcal{S}_{init}\downarrow$ | 1.0 $(\pm 0)$ | 0.20 $(\pm 1.1 \cdot 10^{-2})$ |
| $\|\theta\|_F^2\downarrow$ | $3.1 \cdot 10^3$ $(\pm 3.5 \cdot 10^2)$ | $2.2 \cdot 10^{-5}$ $(\pm 2.5 \cdot 10^{-9})$ |
| $NC1\downarrow$ | $1.2 \cdot 10^5$ $(\pm 7.2 \cdot 10^2)$ | $7.5 \cdot 10^{-14}$ $(\pm 1.5 \cdot 10^{-14})$ |
| $\text{Tr}(\Sigma_W)\downarrow$ | 15.6 $(\pm 1.4)$ | $7.1 \cdot 10^{-18}$ $(\pm 1.1 \cdot 10^{-18})$ |

Table 7: Statistical significance of Table 2

| $\alpha$ | $2 \cdot 10^{-1}$ | $2 \cdot 10^0$ | $2 \cdot 10^1$ |
|---|---|---|---|
| Train Acc.$\uparrow$ | 100 $(\pm 0)\%$ | 100 $(\pm 0)\%$ | 100 $(\pm 0)\%$ |
| Test Acc.$\uparrow$ | 92.7 $(\pm 3.4 \cdot 10^{-2})\%$ | 92.4 $(\pm 5.2 \cdot 10^{-2})\%$ | 88.3 $(\pm 3.7 \cdot 10^{-1})\%$ |
| $\mathcal{D}_{LR}\downarrow$ | **$4.9 \cdot 10^{-2}$** $(\pm 4.4 \cdot 10^{-3})$ | **$1.1 \cdot 10^{-1}$** $(\pm 9.4 \cdot 10^{-3})$ | **$5.6 \cdot 10^{-1}$** $(\pm 6.1 \cdot 10^{-2})$ |
| $\mathcal{S}_{init}\downarrow$ | $6.8 \cdot 10^{-2}$ $(\pm 2.1 \cdot 10^{-3})$ | $4.1 \cdot 10^{-2}$ $(\pm 4.5 \cdot 10^{-3})$ | $5.2 \cdot 10^{-2}$ $(\pm 1.8 \cdot 10^{-2})$ |
| $\|\theta\|_F^2\downarrow$ | $3.4 \cdot 10^3$ $(\pm 1.1 \cdot 10^1)$ | $3.2 \cdot 10^3$ $(\pm 4.4)$ | $3.2 \cdot 10^3$ $(\pm 2.0)$ |
| $NC1\downarrow$ | $2.3 \cdot 10^4$ $(\pm 4.1 \cdot 10^3)$ | $3.2 \cdot 10^3$ $(\pm 4.3 \cdot 10^2)$ | $8.1 \cdot 10^2$ $(\pm 1.4 \cdot 10^2)$ |
| $\text{Tr}(\Sigma_W)\downarrow$ | 2.0 $(\pm 0.25)$ | $3.1 \cdot 10^{-1}$ $(\pm 2.3 \cdot 10^{-2})$ | $1.2 \cdot 10^{-1}$ $(\pm 1.2 \cdot 10^{-2})$ |

## H.2    DISCUSSION ON NC1 MEASURE

In Tables 6 and 7, the NC1 measure is numerically unstable due to the pseudo-inverse of $\Sigma_b$. For stability, we used $\Sigma_b^{\dagger} \approx (\Sigma_b + 10^{-4}I)^{\dagger}$. In addition to NC1, we consider a similar measure $\text{Tr}(\Sigma_W)$ Papyan et al. (2020) for collapse of within class variability, which is also reported in Tables 6 and 7. The measure $\text{Tr}(\Sigma_W)$ also shows a similar trend to NC1.

## H.3    DATASET DETAILS

We use publicly available datasets, including MNIST Deng (2012) and CIFAR-10/100 Krizhevsky et al. (2009) from PyTorch Paszke et al. (2019), and mod $p$ division task from `https://github.com/teddykoker/grokking`.

Table 8: Statistical significance of Table 3

| Task | Architecture | Condition | $\mathcal{D}_{LR}$ |
|---|---|---|---|
| Mod 97 | 2-layer transformer | Step 200 (before grokking) | 0.51 ($\pm$0.04) |
| | | Step 3000 (after grokking) | 0.11 ($\pm$0.01) |
| CIFAR-100 | ResNet18 | learning rate = 0.005 | 0.053 ($\pm$0.01) |
| | | learning rate = 0.05 | 0.025 ($\pm$0.004) |
| | | learning rate = 0.2 | 0.039 ($\pm$0.007) |
| CIFAR-10 | ResNet18 | weight decay = 0 | 0.05 ($\pm$0.003) |
| | | weight decay = $10^{-4}$ | 0.015 ($\pm$0.003) |
| | | weight decay = $10^{-3}$ | 0.003 ($\pm$0.001) |
| CIFAR-10 | ResNet18 | . | 0.026 ($\pm$0.003) |
| | MLP | | 0.48 ($\pm$0.07) |
| CIFAR-10 | ResNet18 | no label shuffling | 0.031 ($\pm$0.004) |
| | | 10% label shuffling | 0.042 ($\pm$0.003) |
| | | full label shuffling | 0.034 ($\pm$0.004) |
| MNIST | CNN | full backpropagation | 0.043 ($\pm$0.005) |
| | | last layer only training | 0.51 ($\pm$0.008) |
| CIFAR-100 | VGG-16 | without batch normalization | 0.66 ($\pm$0.01) |
| | | with batch normalization | 0.073 ($\pm$0.002) |

## H.4 MODEL DETAILS

Our model implementations are based on publicly available code assets, including VGG16 from PyTorch Paszke et al. (2019), ResNet18 from Nakkiran et al. Nakkiran et al. (2021), and the modular division task from `https://github.com/teddykoker/grokking`.

As described in the main text, we only trained our models on MSE loss where the targets are one-hot vectors (up to scaling constant). The constant $\alpha$ is used to scale the output $y \to y/\alpha$.

Table 9: Dataset details

| Figure | dataset | training sample count | $\alpha$ | batch size |
|---|---|---|---|---|
| Fig. 1 | Encoded MNIST (Section H.4.1) | 60,000 (all) | 1/3 | 128 |
| Table 2 | MNIST | 60,000 (all) | 1 | 128 |
| Fig. 2 | CIFAR-10 | . | 1/3 | 128 |
| Fig. 3 | CIFAR-100 | 50,000 (all) | 1/10 | 128 |
| Fig. 5 | CIFAR-100 | 50,000 (all) | 1/10 | 128 |
| Fig. 11 | MNIST | 1,000 | . | 128 |
| Fig. 12 | mod-p division | 4,656 | 1 | 512 |
| Fig. 15 | CIFAR-10 | 10,000 | 1/3 | 128 |

## H.4.1 ENCODED MNIST

In Fig. 1, we encoded the labels as one-hot vectors on the first 10 pixels of the MNIST dataset. For the training set, we encoded the true labels, but for the test set, we encoded random labels.

### H.4.2 CIFAR-10/100

For CIFAR-10/100, we use standard augmentation using random crop 32 with padding 4 and random horizontal flip with probability 0.5. We also use standard normalization with mean $[0.4914, 0.4822, 0.4465]$ and standard deviation $[0.2023, 0.1994, 0.2010]$ for each channel.

Table 10: Training details

| Figure | optimizer | learning rate | momentum / beta | weight decay | epochs | learning rate scheduling |
|---|---|---|---|---|---|---|
| Fig. 1 | Adam | $1 \cdot 10^{-3}$ | $[0.9, 0.999]$ | 0 | 100 | None |
| Table 1 | SGD | $1 \cdot 10^{-6}$ | 0 | $1 \cdot 10^{-3}$ | 200 | None |
| Fig. 2 | SGD | $5 \cdot 10^{-2}$ | 0.9 | $5 \cdot 10^{-5}$ | 200 | $\times 0.2$ per 60 epochs |
| Fig. 3 | SGD | $1 \cdot 10^{-2}$ | 0.9 | $1 \cdot 10^{-5}$ | 400 | None |
| Fig. 5 | SGD | $5 \cdot 10^{-2}$ | 0.9 | $5 \cdot 10^{-5}$ | 400 | $\times 0.2$ per 60 epochs |
| Fig. 11 | Adam | $1 \cdot 10^{-3}$ | $[0.9, 0.999]$ | 0 | 100 | None |
| Fig. 12 | Adam | $1 \cdot 10^{-3}$ | $[0.9, 0.98]$ | 0 | 4000 | None |
| Fig. 15 | SGD | $5 \cdot 10^{-2}$ | 0.9 | $5 \cdot 10^{-5}$ | 200 | $\times 0.2$ per 60 epochs |

### H.5 COMPUTE RESOURCES

The models ran on GPU cluster containing RTX 1080 (8GB), RTX 2080 (8GB), RTX3060 (12 GB), and RTX3090 (24GB). The typical time to train a model is 2 hours, but it varies from 10 minutes to 6 hours depending on the experiment. The evaluation metrics take less than 5 minutes and may need up to 2GB of CPU memory.

### H.6 USE OF LLMS

Large language models (LLMs) were used to polish writing to make the paragraphs more concise.

# I INTERMEDIATE LAYER FEATURES

Because our paper focuses on a practical measure of dynamical richness rather than explaining the rich dynamics themselves, we restricted our study to the computationally efficient last-layer features. Nevertheless, to encourage future work on rich dynamics using our approach, we include preliminary explorations of intermediate layers.

We can generalize our metric and visualization by replacing $\mathcal{T}$ in Eq. (1) with that of the $l^{\text{th}}$ layer:

$$\mathcal{T}^{(l)} := \sum_{k=1}^{p_l} |\Phi_k^{(l)}\rangle \langle \Phi_k^{(l)}|, \tag{59}$$

where $\Phi^{(l)}$ is the $l^{\text{th}}$ layer's feature map and $p_l$ is its width. When the output of the feature map $\Phi^{(l)}$ is multidimensional (e.g. the output from a 2D convolutional module), the output is flattened to a vector (similarly to Montavon et al. (2011); Alain & Bengio (2016); Raghu et al. (2017)).

However, several challenges arise when considering features from earlier layers. First, the inner product (correlation) between intermediate-layer features no longer reflects their alignment with the output (or with $\mathcal{T}_{MP}$), since these features undergo nonlinear transformations before reaching the last layer. This limits the interpretability of CKA, $\Pi^*$, and $\hat{\Pi}$, all of which rely on feature inner products. Still, we can predict how linearly correlated these features are with the output or $\mathcal{T}_{MP}$.

Second, computation becomes more expensive. Whereas last-layer features are typically narrow ($\sim 10^3$), intermediate layers are often much wider, and our methods scale quadratically with width. Larger width also leads to finite-size effect from the dataset, which we discuss in Section J.

With these caveats in mind, we present the behavior of intermediate-layer features. Because the metric lacks a clear theoretical interpretation in this setting, we provide only visualizations. Fig. 18 shows ResNet18 trained on CIFAR100, which exhibits rich dynamics under our metric (Table 3). While we observe a depth-wise transition from using many weakly aligned features to using fewer but more aligned ones, it is difficult to discern lazy/rich dynamics from the intermediate layers alone. The effect of learning rate is shown in Fig. 19, but the intermediate layers again exhibit no additional information compared to the last layer.

Fig. 20, corresponding to Fig. 3, shows the lazy-to-rich transition in VGG16 with and without batch normalization. With batch normalization, the model follows the use of many weakly aligned features to the use of fewer but more aligned ones in depth. Without batch normalization, we observe no comparable shift. Moreover, for the model without batch normalization, layers 3–6 show little variation, suggesting that the convolutional layers learn minimal feature structure and that training primarily affects the final fully connected layers only. Yet, the difference between lazy and rich dynamics is most pronounced for the last layer.

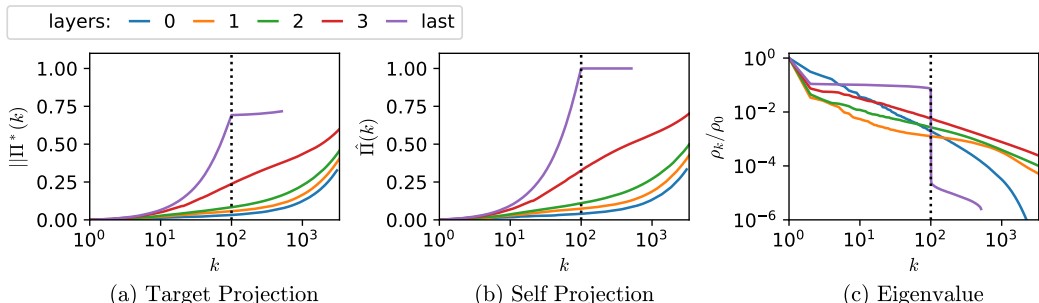

(a) Target Projection     (b) Self Projection     (c) Eigenvalue

Figure 18: **ResNet18 on CIFAR100.** Colors represent the features of the intermediate and the last layer. See Fig. 22 for the definition of the layers. As the layers go deeper, we observe transition from many less correlated features to fewer more correlated features. The purple line ends early as the last layer only has 512 features.

As shown in Figs. 19 and 20, the low-rank bias is most pronounced in the last layer, enabling our computationally efficient richness measure in the main text. This motivates the hypothesis that the

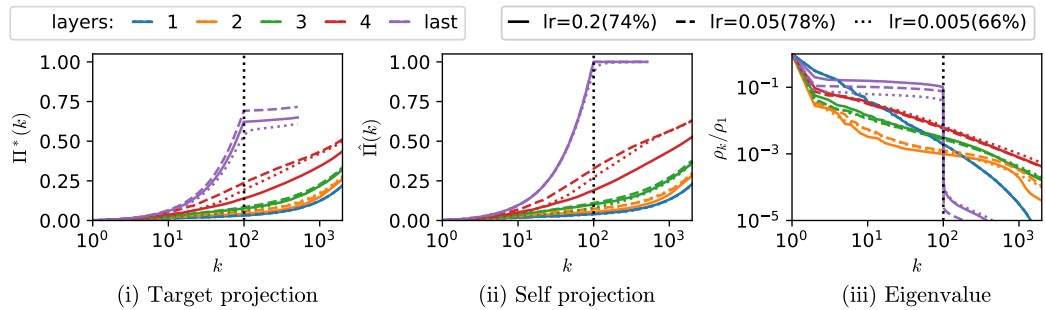

Figure 19: **Effect of learning rate for ResNet18 on CIFAR100.** The models corresponds to Fig. 4.

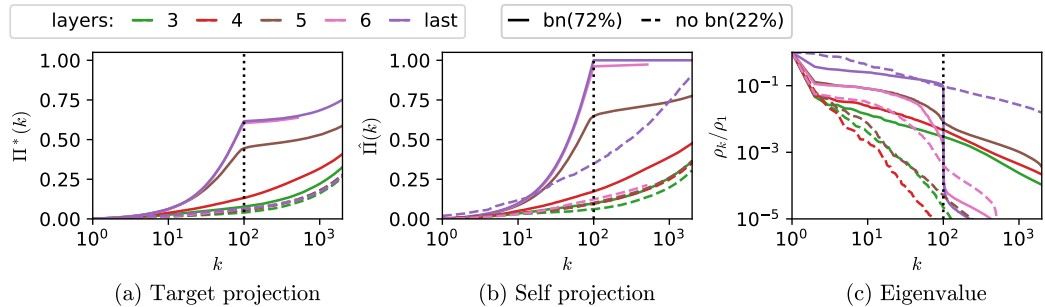

Figure 20: **Effect of batch normalization in VGG trained on CIFAR100**. This is the intermediate values for Fig. 3. Solid and dashed line indicate the model with and without batch normalization, respectively.

last layer consistently exhibits the strongest low-rank bias. To test this, we examined the greedy layerwise training method (Ivakhnenko & Lapa, 1965), which trains each layer by appending an auxiliary classifier, freezing the layer after training, and then repeating the process for subsequent layers.

Fig. 21 shows that greedy layerwise training, compared to full backpropagation, causes feature collapse in earlier layers (e.g., layer 2), indicating that the last layer carries the strongest low-rank inductive bias. This supports our focus on last-layer features for lazy/rich detection. Under greedy layerwise training, feature quality also stagnates after layer 2, leading to poorer overall performance. This observation suggests that such early collapse dynamics may be suboptimal, warranting further theoretical study.

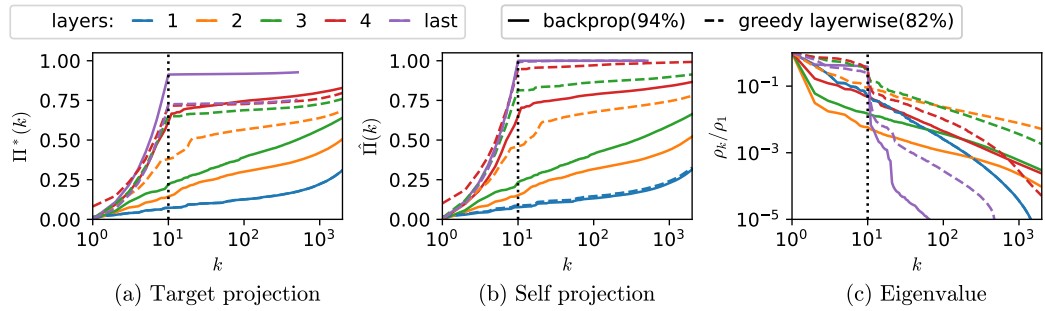

Figure 21: **Greedy layerwise training leads to a collapse of features in earlier layers.** In contrast to full backpropagation (solid), greedy layerwise training (dashed) leads to more pronounced alignment at lower $k$ for earlier layers.

## I.1 LAYER DETAILS

Fig. 22 shows the layers defined in this section. The notation $k \times k$ Conv$(\cdot, \cdot, \cdot, \cdot)$ refers to the 2d convolutional module with a kernel of size $k \times k$, and arguments within parenthesis refer to the input channel, output channel, stride, and padding respectively. Linear$(\cdot, \cdot)$ refers to the fully connected module with the arguments referring to input width and output width respectively. MaxPool$(\cdot, \cdot, \cdot)$ refers to max pooling with the arguments as kernel size, stride, and padding respectively. Adaptive Avgpool$(\cdot, \cdot)$ refers to adaptive average pooling where arguments are the size of the image after pooling. The curved arrow refers to the skip connection (identity), and any modules that appear in the arrow are applied. BN refers to batch normalization and (BN, ReLU) refers to the application of batch normalization and ReLU in respective order.

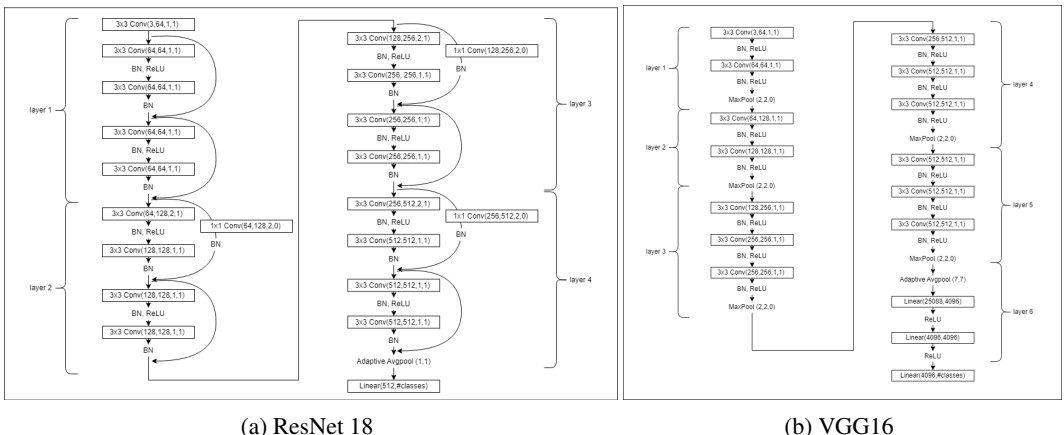

(a) ResNet 18          (b) VGG16

Figure 22: **Architectures and layers** The diagrams show the CNN architectures used in the study and their layers. **(a):** ResNet18 architecture. The size of feature space (post-activation) for each layer is 65536, 32768, 16384, and 512 respectively. **(b):** VGG16 architecture. The feature space size for each layer is 16384, 8192, 4096, 2048, 512, and 4096. Note that the penultimate layer only consists of linear modules (fully connected) and non-linearities.

# J  ACCURACY OF APPROXIMATION METHODS

In Section 5.1, we proposed three vector metrics for visualization: quality $Q^*$, utilization $\hat{Q}$, and eigenvalues (or intensity) $\rho$. These measures require the true input distribution $q$, which we cannot access in practice. The approximation method for these measures was described in Section E – the Nyström method Baker & Taylor (1979) for approximating $e_k$. In this section, we provide an empirical example – where the true eigenfunctions and eigenvalues are known – to assess the validity of Nyström method in approximating eigenvalues and eigenfunctions. We also discuss why quality and utility were presented in a cumulative manner ($\Pi^*$ and $\hat{\Pi}$) while the eigenvalues ($\rho_k$) are presented directly.

## J.1  THE NYSTRÖM METHOD

As discussed in Section E, the Nyström method diagonalizes the empirical feature matrix $\Phi(X) \in \mathbb{R}^{p \times n}$ to approximate the eigenfunctions $e_k$ and eigenvalues $\rho_k$. This is more explicitly written in lines 6 and 7 of Algorithm 1:

$$\rho_k \leftarrow s_k^2, \qquad e_k(x) \leftarrow \Phi(x)^T u_k / s_k, \tag{60}$$

where $u$ and $s$ are the left orthogonal matrix and singular values obtained from singular value decomposition of $\Phi(X)$.

### J.1.1  EXPERIMENT SETUP FOR TESTING NYSTRÖM METHOD

The accuracy of the Nyström method depends on the accuracy of singular values ($s_k$) and left eigenvectors ($u_k$) of a random matrix ($\Phi(X)$). To empirically assess the accuracy of the method, we will use $p$ independent Gaussian random variables to sample our feature matrix. A power-law distribution with exponent $\alpha$ will be used as the eigenvalues:

$$\Phi(x) \sim \mathcal{N}(\vec{0}, Diag([\rho_1, \rho_2, \cdots, \rho_p])), \qquad \rho_k = k^{-\alpha}. \tag{61}$$

Gaussian random variables serve as valid features since the features are random variables ($x \sim q$ is a random variable, and $e_k$ is a fixed map, thus $e_k(x)$, $x \sim q$ is also a random variable) and are orthogonal to other eigenfunctions:

$$e_k(x) := \frac{1}{\sqrt{\rho_k}}\Phi_k(x), \qquad \langle e_i | e_j \rangle = \delta_{ij}. \tag{62}$$

Eigenfunctions can also be expressed with $o_k$ as

$$e_k(x) = \Phi(x)^T o_k / \sqrt{\rho_k} \tag{63}$$

where $o_k$ is one-hot vector with its $k^{th}$ entry equal to 1 and all other entries 0 (i.e. $o_k = [0, \cdots, 0, 1, 0, \cdots, 0]$). Indeed we can check that $e_k$ is the eigenfunction of $T$ with eigenvalue $\rho_k$

$$T\left[\frac{\Phi(x)^T o_k}{\sqrt{\rho_k}}\right](x') = \int \Phi(x')^T \Phi(x) \frac{\Phi(x)^T o_k}{\sqrt{\rho_k}} q(x) dx \tag{64}$$

$$= \sqrt{\rho_k}\Phi(x')^T o_k \tag{65}$$

$$= \rho_k e_k, \tag{66}$$

where in the second line, we used that $o_k^T$ is a one-hot vector and all entries of $\Phi$ are independent.

### J.1.2  ACCURACY MEASURES

With the underlying true features and eigenvalues defined, we propose measures for calculating the accuracy of approximated values obtained via Algorithm 1.

The error of eigenvalue is measured as square distance between $\rho_k$ (true value) and $s_k^2(X)$ (approximation) normalized by $\rho_k$:

$$\mathbf{E}_X\left[\frac{(s_k^2(X) - \rho_k)^2}{\rho_k^2}\right], \tag{67}$$

Note that $\mathbf{E}_X$ is expectation over all possible features matrices $\Phi(X)$.

The error of eigenfunction is measured as 1 minus the inner product between $u_k(X) \in \mathbb{R}^p$ and $o_k$:

$$1 - \mathbf{E}_X \left[ u_k^T o_k \right] . \tag{68}$$

The inner product $u_k^T o_k$ is proportional to the inner product between $e_k^{approx}$ and $e_k$:

$$\langle e_k^{approx}, e_k \rangle = \int \left( \frac{\Phi(x)^T u_k(X)}{s_k(X)} \right)^T \frac{\Phi(x)^T o_k}{\sqrt{\rho_k}} q(x) dx \tag{69}$$

$$= \frac{u_k(X)^T}{s_k(X)} \int \Phi(x) \Phi(x)^T q(x) dx \frac{o_k}{\sqrt{\rho_k}} \tag{70}$$

$$= \frac{u_k(X)^T}{s_k(X)} \mathrm{Diag}(\rho_1, \ldots, \rho_p) \frac{o_k}{\sqrt{\rho_k}} \tag{71}$$

$$= \frac{u_k(X)^T o_k}{s_k(X)\sqrt{\rho_k}}, \tag{72}$$

where the denominator is not of interest as we are not particularly interested in the norm of the $e_k^{approx}$.

### J.1.3 RESULTS

In Fig. 23 we generate a feature matrix containing $n = 10,000$ datapoints and $p = 1000$ features with Gaussian random variables. We use the Nyström method to approximate the eigenvalues and eigenfunctions and assess their accuracy with the metrics in Eq. (67) and Eq. (68). We observe in Fig. 23(a) that eigenvalues are approximated precisely except for eigenvalues with $k \approx p$. In Fig. 23(b), we observe that the accuracy of the eigenfunctions correlates with the respective eigenvalues: the larger the eigenvalue (compared to the median eigenvalue), the better the accuracy. In the main text, we are often interested only in the first few eigenfunctions with large eigenvalues; we can conclude that the Nyström method is sufficiently accurate for our needs.

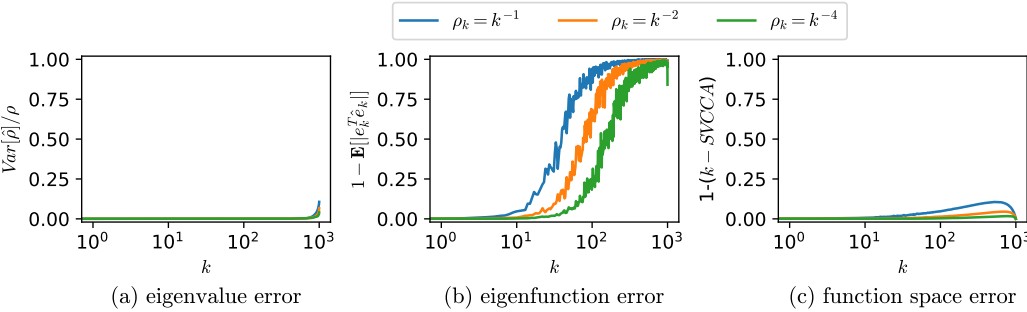

(a) eigenvalue error      (b) eigenfunction error      (c) function space error

Figure 23: **Nyström approximation for different eigenvalue distributions** A feature matrix was sampled from Gaussian random variables and then approximated with Nyström method. Power law eigenvalue spectra with different decaying exponents were tested. (a) The eigenvalues are accurately predicted except for the tail eigenvalues with $k \approx p$. (b) The eigenfunctions are accurate only when the eigenvalues are large compared to the median eigenvalue. (c) The function space error below $0.25$ for all $k$.

### J.2 FUNCTION SPACE APPROXIMATION

The Nyström method's prediction accuracy degrades for smaller eigenvalues. The spanned space of $[e_1^{approx}, \cdots, e_k^{approx}]$, however, often accurately approximates that of $[e_1, \cdots, e_k]$; the larger error of the $e_k^{approx}$ is often due to the cumulation of error in predicting $e_i^{approx}$ for $i < k$. Fig. 24 illustrates an example where $[u_1, u_2, u_3]$ span the same space as $[o_1, o_2, o_3]$, and $u_1$ and $u_2$ (thus $e_1$ and $e_2$) are well approximated – the errors in predicting $u_1$ and $u_2$ accumulate to the error of predicting $u_3$, and results in a poorer approximation for $u_3$.

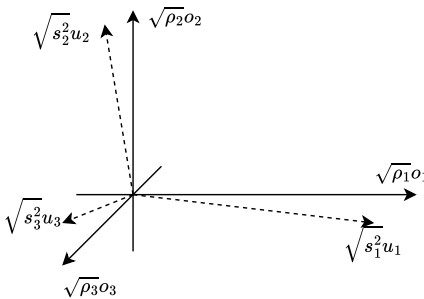

Figure 24: **Approximation on different eigenvalue distributions**

We can use the Gaussian random variable setup to check the error of the function space approximation. To measure the similarity between $\text{span}([e_1, \cdots, e_k])$ and approximated function space $[e_1^{approx}, \cdots, e_k^{approx}]$, we can use k-SVCCA (Raghu et al., 2017) or the cumulative projection measure in Eq. (5).

$$\frac{1}{k} \sum_{i=1}^{k} \sum_{j=1}^{k} \langle e_i | e_j^{approx} \rangle^2 . \tag{73}$$

Since we can express the inner product between true and approximated eigenfunctions with $u_k$ and $o_k$ (Eq. (69)), we can express the metric in terms of $u_k$ and $o_k$:

$$1 - \frac{1}{k} \sum_{i=1}^{k} \sum_{j=1}^{k} \left( u_i^T o_j \right)^2 , \tag{74}$$

where we subtracted the measure from 1 so the measure will be 0 when there is no error. In Fig. 23(c), we observe that the function space can be approximated more accurately in comparison to individual eigenfunctions in Fig. 23(b); This is why we often present the quality and utility in cumulative measure $\Pi^*$ or $\hat{\Pi}$.

### J.3 ERROR IN PROJECTION MEASURES

We now discuss the error in the projection measures like $\Pi^*$. The projection measure (Eq. (5)) consists of the sum of squares of inner products between eigenfunctions and function of interest (target or expressed). The Riemann sum over the **test set** $S_{te}$ approximates the inner product – similar to how test loss approximates generalization loss.

$$\frac{\langle e_k, f_j \rangle^2}{\|e_k\|_2^2 \|f_j\|_2^2} = \frac{\left( \int_{\mathcal{X}} e_k(x) f_j(x) q(x) dx \right)^2}{\int_{\mathcal{X}} e_k(x)^2 q(x) dx \int_{\mathcal{X}} f_j(x)^2 q(x) dx} \approx \frac{\left( \sum_{x^{(i)} \in S_{te}} e_k(x^{(i)}) f_j(x^{(i)}) \right)^2}{\sum_{x^{(i)} \in S_{te}} e_k(x^{(i)})^2 \sum_{x^{(i)} \in S_{te}} f_j(x^{(i)})^2} . \tag{75}$$

Using the Riemann sum, we can measure the correlation between spanned spaces (Eq. (5)). Using the vector representation of eigenfunction in the test set $e_i^{approx}(X_{test}) \in \mathbb{R}^{n_{test}}$ and the vector representation of function of interest $f_i(X_{test})$, we can approximate the quantities of Eq. (5) as

$$\sum_{j=1}^{C} \sum_{i=1}^{p} \langle e_i | f_j \rangle^2 \approx \text{Tr}(EF^T FE^T), \quad E_{ij} = \frac{e_i^{approx}(x^{(j)})}{\sum_j^n E_{ij}^2}, \quad F_{ij} = \frac{f_i(x^{(j)})}{\sum_j^n F_{ij}^2}. \tag{76}$$

where $x^{(j)}$ are iterated over the test set. If the rows of $E \in \mathbb{R}^{p \times n_{test}}$ and $F \in \mathbb{R}^{C \times n_{test}}$ are orthogonal, then Eq. (76) is smaller than equal to 1: consistent with the definition of Eq. (5).

However, Eq. (76) is susceptible to error when: rows of $E$ are not orthogonal and when $p \approx n_{test}$. We will discuss how we handled such problems in the paper.

#### J.3.1 NON ORTHOGONAL EIGENFUNCTIONS

As discussed earlier, approximated eigenfunctions are less accurate when their eigenvalues are smaller; they may not even be orthogonal in $q$ or the empirical distribution of the test set. The inner

product between two approximated eigenfunctions is

$$\langle e_i^{approx} | e_j^{approx} \rangle = \frac{1}{s_i s_j} \int_{\mathcal{X}} u_i^T \Phi(x) \Phi^T(x) u_j q(x) dx \tag{77}$$

$$= \frac{1}{s_i s_j} u_i^T \left( \int_{\mathcal{X}} \Phi(x) \Phi^T(x) q(x) dx \right) u_j \tag{78}$$

$$= \frac{1}{s_i s_j} u_i^T J u_j, \tag{79}$$

where the Fisher information matrix $J \in \mathbb{R}^{p \times p}$ (Karakida et al., 2021) is defined as $\int_{\mathcal{X}} \Phi(x) \Phi^T(x) q(x) dx$. Eq. (77) equals Kronecker delta if only if $[u_1, \cdots, u_p]$ are the eigenvectors of $J$ with eigenvalues $[s_1^2, \cdots, s_p^2]$. Since we obtained $u_i$ from the empirical feature matrix of the training set, the approximated eigenfunctions, especially those with small eigenvalues, are often not orthogonal in $q$ (and in the test set).

To handle such an issue, we use QR decomposition on $E = [e_1^{approx}(X_{test}), \cdots, e_p^{approx}(X_{test})] \in \mathbb{R}^{p \times n_{test}}$ before applying Eq. (76): because QR decomposition orthogonalizes the rows; preserves the span of the first $k$ vectors; handles overlapping vector between rows by conserving larger eigenvalue eigenvectors (eigenfunctions) - which are more accurate.

In Fig. 25, we plot the cumulative measure with and without QR decomposition for ResNet18 on CIFAR10. We observe that $\Pi^*$ and $\hat{\Pi}$ grow beyond 1 without QR decomposition, indicating that approximated eigenfunctions are not orthogonal; The inconsistent behaviour is removed when QR decomposition is applied. Note that the deviation between QR and no-QR plots emerges when the eigenvalues are sufficiently small (larger error in approximating the eigenfunction).

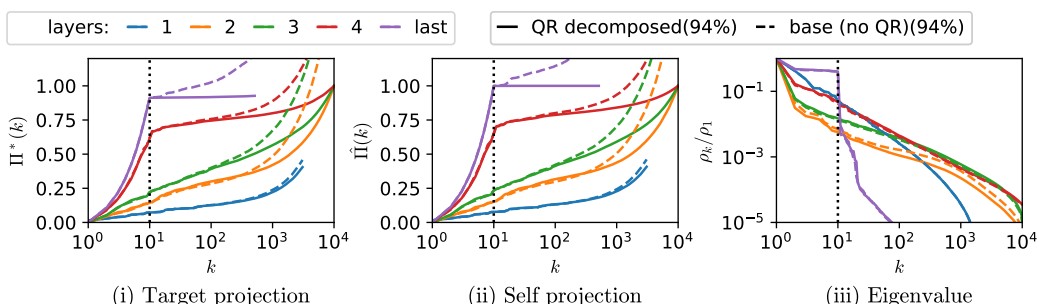

(i) Target projection  (ii) Self projection  (iii) Eigenvalue

Figure 25: **Comparing approximation methods for the projection measures.** ResNet18 on CIFAR10. For (a) and (b), the measures are calculated with (solid) and without (dashed) QR-decomposition. Note that QR decomposition removes the inconsistent behavior of the cumulative projection measures (e.g. larger than 1). With QR decomposition, the projection measure for layers with $p \geq 10^4$ consistently equals 1 when $k = n_{test} = 10^4$.

### J.3.2 FINITE SIZE EFFECT

For two sets of basis functions, increasing the number of bases to a large but finite value does not guarantee any overlap between the two function spaces: because functions are infinite-dimensional vectors. This is not the case for two finite-dimensional matrices: If $p = n_{test}$ in Eq. (76) and $E$ has full rank, then any $F$ will be spanned by the rows of $E$.

In Fig. 25(a,b), we observe that $\Pi^*(n_{test})$ and $\hat{\Pi}(n_{test})$ are equal to 1 for layers with $p > n_{test}$. This does not indicate that intermediate features can express the target/expressed function; It is achieved because $n_{test}$ orthogonal vectors ($E$) can always span $n_{test}$ dimensional vector space (and thus the row space of $F$).

