# OpenReview forum: "Decoupling Dynamical Richness from Representation Learning: Towards Practical Measurement"
_ICLR.cc/2026/Conference — ICLR 2026 Poster_

### Official Review · Reviewer_cwga · 2025-10-27

**Soundness:** 4
**Presentation:** 4
**Contribution:** 3
**Rating:** 8
**Confidence:** 3

**Summary:**

This paper introduces a new metric for quantifying rich (feature) learning. The authors motivate this by distinguishing between two perspectives on feature learning: a representation perspective (focus on usefulness of features for performance/generalization) and a dynamics perspective (focus on the transformation of features). They show that although it is often associated with representational usefulness, rich learning doesn’t necessarily lead to generalization and instead reflects an inductive bias towards certain solutions. Thus, the paper focuses on dynamic feature learning, introducing a computationally-efficient and performance-independent metric that effectively compares the network activations before and after the last layer. They show that this metric generalizes neural collapse, a phenomenon associated with rich learning. By comparing to prior measures of rich dynamics, they illustrate special cases where their metric captures dynamical richness, while others do not. They also conduct experiments across several different models and datasets, demonstrating how different training conditions affect performance and richness separately. Finally, the authors provide complementary visualization methods, make it easier to see the low-rank feature bias of rich dynamics.

**Strengths:**

The presentation and clarity of the paper is excellent. It is well-written, clear, and accessible. The paper is well-motivated, with a comprehensive introduction and background.

The soundness of the paper is also excellent. The authors take care to support each of their claims with ample evidence and solid experiments/methodology. The paper identifies an important distinction in perspectives on feature learning (dynamics- versus representation-focused) that challenges implicit assumptions about rich learning necessarily leading to improved performance/generalization and provides experiments to support this. The metric introduced has good potential for studying feature learning, as it is efficient to compute and captures rich dynamics in settings where prior metrics fail. The authors provide well-thought experiments to demonstrate the utility of their metric and support their central claims, showing that it captures grokking, neural collapse, and other known feature learning phenomenon. They also provide helpful visualization methods, which they show the utility of in interpreting different network behavior. The paper offers a good contribution to the ICLR community and I recommend it for acceptance. It was a pleasure to read and I thank the authors for their solid work.

**Weaknesses:**

To clarify, I think these points are relatively specific (possibly only applying to certain edge cases) and that the work as a whole still stands well despite some potential limitations.

One potential weakness of the metric introduced is that it’s limited in its generality (to orthogonal and isotropic target functions). However, this is properly acknowledged by the authors and the metric still captures many classification tasks.

As the authors also state, their metric depends on a comparison between the last two layers. Thus, two networks with different feature learning behavior in earlier layers may appear identical if their final two layers are similar. I’m wondering if there would be any principled way to applying the same approach to other layers in the network? And are there concrete settings where the comparison between the last two layers is a major limiting assumption?

As far as I understand, the metric is based on the low-rank bias of rich dynamics. I know that this might be rare in practice, but would the metric still work if the target function is full-rank? Is the assumption of low-rank dynamics potentially limiting?

The authors state that in Figure 5 “feature quality correlates with feature intensity during training, with larger features improving faster… the correlation between quality and intensity during training has not been previously observed or studied.” Isn’t this expected, as it’s known that large features (singular values) are learned first (Saxe et al., 2014)? Or, do you mean that eigenvector alignment is occurring simultaneously with singular vector scaling? In Atanasov et al., 2022, they show that for anisotropic data, “the NTK must necessarily change its eigenvectors when the loss is significantly decreasing, destroying the silent alignment phenomenon.” I’m a bit skeptical of the statement that this correlation has not been observed or studied.

**Questions:**

1. Would the metric still work if the target function is full-rank?
2. Would be any principled way to applying the same approach to other layers in the network? And are there concrete settings where the comparison between the last two layers is a limiting assumption?

Suggestion: there are a few typos but nothing major

---

> ### Author Response · Authors · 2025-11-19
>
> We deeply thank the reviewer for a kind and thoughtful appreciation of our paper.
>
> ### Response to Weaknesses
>
> * **About the generality of the metric...** Our theoretical results are indeed derived under orthogonal and isotropic target assumptions, which are standards in NC literature. We have discussed this further in our response to Common Comments 1.
>
> * **About intermediate layers...** We added a new Appendix I detailing our exploration of intermediate layers (please see Common Comment 2). Figures 18, 20, and 21 suggest that the last layer exhibits the strongest low-rank bias and that richness is most pronounced there. We suspect that certain extreme settings (e.g., very deep, large early-layer initialization, small later layer initialization, strong weight decay, and limited data) may yield cases where the last two layers are uninformative, though we could not reproduce a clear example. We believe such cases are rare in practice.
>
> * **About full-rank...** For clarification, we interpret "full-rank" in your question as the case where the width of the last layer is the same dimension as the output. This is an interesting question we have not explored. It will be difficult to see a significant difference, but we expect to see some $D_{LR}$ differences coming from the eigenvalue distribution based on our observations on lazy/rich dynamics. In lazy dynamics, the features are more or less frozen and their intensities often follow a power-law (See for example, blue line in Fig. 16(iii)). This is in contrast to the typical rich dynamic where the intensities follow a uniform distribution up to $C$ (See for example, orange line in Fig. 16(iii)).
>
> * **About the comment regarding Figure 5…** We agree that the correlation between intensity and learning time offers useful intuition. However, the setup of Saxe et al. (2014) does not involve changes in alignment (i.e., input–output correlations of singular directions remain fixed), and intensity–training-speed correlations also arise in kernel settings (Appendix C). Our intended point was the latter: that eigenvector alignment occurs simultaneously with singular-value scaling. We agree that the original wording may have been misleading, and fixed comments relating to Fig. 5.
>
>     What we aimed to highlight is that the strong correlation between singular-vector scale and alignment is an unexpected phenomenon that does not follow from the silent alignment effect. Although Atanasov et al. (2022) discussed the conditions under which silent alignment weakens (e.g., anisotropic inputs, non-linear networks), they concluded that “alignment with the target function varies in a non-trivial way (for NTK)” in more practical settings, rather than suggesting a strong strength–alignment correlation such as that in Fig. 5. To the best of our knowledge, this strong correlation in the last layer features during training has not been studied.
>
>     Thank you for a sharp spotting.
>
>
> ### Response to Questions
> 1. Please see our comment on “About full-rank” in the Response to Weakness above.
>
> 2. Please see our comment on “About intermediate layers” and common comments 2.
>
>
> ### Response to Suggestions:
>
> Thank you, we have fixed typos in the revised version.
>
> ### References:
>
> Saxe et al. (2014) and Atanasov et al. (2022) are cited in the original submission of the manuscript.

---

> > ### Comment · Reviewer_cwga · 2025-11-27
> > **Rebuttal Acknowledgement**
> >
> > Thank you to the authors for their thorough rebuttal. I appreciate the additional analysis of intermediate layers and clarifications provided. All of my comments have been addressed. I echo Reviewer GFrJ that this is a solid and impactful submission and I maintain my score, recommending an accept.

---

> > > ### Author Response · Authors · 2025-11-28
> > >
> > > We deeply thank the reviewer for the valuable feedback and for recognizing the improvements in our revision. We truly appreciate the time and effort invested in the review process, which we believe has strengthened our paper.

---

### Official Review · Reviewer_Czxk · 2025-10-29

**Soundness:** 2
**Presentation:** 2
**Contribution:** 2
**Rating:** 4
**Confidence:** 3

**Summary:**

The paper proposes a performance-independent metric for dynamical richness (DLR) that compares the feature kernel built from penultimate-layer activations with a “minimum projection” operator determined by the network’s learned function. Intuitively, truly rich dynamics should learn only the minimal features needed to span the learned function space; DLR quantifies the gap to this ideal and is normalized in [0,1] (lower is richer). The authors show that when the feature operator attains this minimum form, neural collapse conditions follow as a special case, linking the metric to established phenomena. Empirically, DLR is lightweight to compute and tracks lazy-to-rich transitions across MLPs/CNNs/Transformers, and an eigendecomposition-based visualization complements the scalar score.

**Strengths:**

1. Originality. Recasts richness as low-rank alignment between features and the learned function via a principled MP-operator; this is a fresh angle relative to NTK-deviation or label-based collapse metrics. The reduction to neural collapse provides a clean conceptual bridge.
2. Quality. The metric is simple, normalized, and computationally cheap, enabling use on standard vision models. Comparative experiments are thoughtful and reveal expected patterns.
3. Clarity. The paper is well structured and the three-panel eigendecomposition plots (quality/utilization/eigenvalues) are helpful.
4. Significance. A lightweight, performance-independent diagnostic for dynamics could become a standard tool, much as CKA did for representational comparisons.

**Weaknesses:**

1. DLR inspects only the final-layer features; rich dynamics might manifest earlier and be attenuated by a constrained head. Consider a hierarchical variant (layer-wise DLR or block-DLR) and show whether conclusions persist will be helpful.
2. The technical background is not much sufficient for readers to capture the essence. More straightforward interpretation and introduction are helpful.
3. The MP-operator and some guarantees rely on orthogonal/isotropic targets and supervised, one-hot settings; this narrows immediate applicability (e.g., class imbalance, multilabel, regression, self-supervised).
4. Broader settings (imbalance/SSL), deeper ablations (layer-wise, NTK-feasible baselines), and stronger analysis of assumptions would raise confidence and impact.
5. Writing/format polish. Minor issues distract: e.g., “batch nomralization” typo in Table 3; occasional spacing/notation inconsistencies around figures/equations; some insufficient definitions/clarifications on heavy-weight mathematical notations..

**Questions:**

1. Can DLR be generalized to non-isotropic/imbalanced tasks (e.g., via class-reweighted inner products or whitening), and does the neural-collapse link still hold?
2. What happens if you compute DLR per layer (or per block) and aggregate? This could localize where richness emerges and inform architecture design.
3. Can you theoretically connect decreasing DLR with transition conditions in lazy-to-rich analyses/grokking, beyond empirical alignment?
4. Why CKA over alternatives (e.g., centered HSIC variants)? Any cases where CKA’s centering removes signal that matters for DLR?

**Details Of Ethics Concerns:**

No ethics concern.

---

> ### Author Response · Authors · 2025-11-19
>
> We appreciate the reviewer’s positive remarks regarding the strengths of our work and for taking the time to review our paper.
> ### Response to weaknesses
> 1. In the revised version, we added Appendix I, exploring the intermediate layers. We also did not find low-rank bias manifesting in the earlier layers. Please see 2 of our common comments.
> 2. Please see 3 of our common comments. Could the reviewer specify the technical background that was difficult to access/understand? We are happy to improve the relevant content (e.g. Appendix B).
> 3. Balanced classification is a common assumption in NC literature, with generalization (e.g. imbalanced classification) still an active area of research. Additionally, lazy/rich dynamics typically focus on supervised setups only. Please see 1 of our common comments.
> 4. Please see 1 and 2 of our common comments. The scope is not limiting for the subfield interested in the lazy/rich dynamics or neural collapse.
> 5. Thank you, we fixed the typos in the revised version. Can the reviewer kindly specify notations or definitions that were difficult to understand? We are happy to fill in any missing notations in the glossary (Appendix A).
> ### Response to Questions
> 1. Please see 1 of the common comments.
> 2. Our exploratory results on intermediate layers are now in Appendix I. Please see 2 of the common comments.
> 3. The connection between low-rank bias and rich dynamics for two layer linear networks is given in Saxe et al., (2014) and Mixon et al., (2020). See Nam et al., (2025) for a comprehensive overview. Current theories lack predictive power even at a scale of ReLU MLP on CIFAR-10. Our motivation was to reduce the gap between theory and practice by proposing a metric that may help us interpret phenomena more clearly. A full theoretical treatment of this connection is an interesting direction for future work and would require dedicated papers.
> 4. CKA is normalized and centered HSIC (Kornblith et al. 2019). We wish to have a normalized metric, so normalized and centered HSIC (or its variant) would lead back to CKA or similar variants. While other alignments may exist, CKA is a standard metric used in the kernel literature for its connection to generalization (Appendix C). It also can be decomposed into the basis of our visualization (Appendix C), suitable for a consistent presentation. Centering has been empirically effective for comparing neural network features (Kornblith et al., 2019), as it removes uninformative constant components (Loosely speaking, $a_1$ component in Def. 1, see Appendix B.3). Additionally, there is an analogy between the centering operation and the global mean vector in neural collapse (Appendix D.2).
>
> ### References:
>
> Saxe et al., (2014); Mixon et al., (2020); Nam et al., (2025); Kornblith et al., (2019) are cited in the original submission of the manuscript.

---

> > ### Comment · Reviewer_Czxk · 2025-11-21
> >
> > Thanks for the responses and revised manuscript. I find that the overall contribution is non-trivial, thus I plan to raise my score from 4 to 6 (weak accept). However, I still feel that some of my original concerns are only partially addressed, e.g., the metric still fundamentally relies on orthogonal/isotropic targets and balanced supervised classification, the new intermediate-layer results in Appendix I are rather preliminary and limited in scope. These limitations are now more clearly acknowledged in the discussion, but they still leave open how broadly $D_{LR}$ can be applied beyond standard, balanced classification settings and how well it captures network-wide dynamics.
> > Therefore, I do not think the paper yet reaches the level of an 8 (“good paper”), mainly because the scope of validated settings is still narrow, and the theoretical connection to existing methods remains mostly qualitative, and the contents are still heavier for a broad ICLR audience.

---

> > > ### Author Response · Authors · 2025-11-21
> > >
> > > We deeply appreciate the reviewer’s time and effort in participating in the review process, as well as the decision to increase the score.

---

### Official Review · Reviewer_ztAm · 2025-10-30

**Soundness:** 3
**Presentation:** 2
**Contribution:** 2
**Rating:** 4
**Confidence:** 3

**Summary:**

This paper proposes DLR (Dynamical Low-Rank measure), a computationally efficient and performance-independent metric for quantifying dynamical richness in neural networks. The metric is grounded in the low-rank bias characteristic of rich dynamics and compares activations before and after the last layer via a functional kernel operator. The authors show that DLR reduces to neural collapse as a special case and empirically validate that it captures known lazy-to-rich transitions (e.g., grokking) without relying on accuracy. They further introduce an eigendecomposition-based visualization tool to enhance interpretability.

**Strengths:**

- Important problem: Decoupling dynamical richness from representation quality addresses a fundamental issue in understanding neural network training. The observation that rich dynamics ≠ better generalization (Figure 1) is compelling.
- Computationally efficient: The O(p²C) complexity is a significant improvement over NTK-based methods, making the metric practical for modern architectures.
- Strong theoretical grounding: The connection to neural collapse (Propositions 1 and 2) provides solid theoretical foundation, while extending applicability beyond classification (e.g., regression with scalar output).
- Performance-independent: Unlike existing metrics (Sinit, parameter norm, NC1), DLR doesn't rely on accuracy, initial kernel, or class labels, making it more robust (Tables 1 and 2).

**Weaknesses:**

- Limited scope: The current formulation only applies to orthogonal and isotropic target functions. While this covers many classification tasks, the restriction is significant. The authors acknowledge this but don't provide a clear path to generalization.
- Empirical validation: While the authors demonstrate DLR's utility across diverse settings (grokking, learning rate variations, weight decay, batch norm) the experiments conducted are of small scale and somewhat artificial. It would be interesting to see this applied to more modern scenarios, i.e. bigger models and more difficult datasets.
- Last-layer focus: By only examining last-layer features, the metric misses dynamics in earlier layers. While this is a trade-off for efficiency, it's unclear how much information is lost.
- Limited theoretical analysis: Beyond the connection to neural collapse, there's limited theoretical characterization of when DLR accurately reflects "richness" or what DLR values imply about learning.
- Comparisons could be broader: NTK-based measures are set aside due to computational cost; although justified, a scaled-down NTK comparison on smaller models (which the authors already use) would strengthen claims that DLR is a better proxy rather than merely cheaper.
- Clarity: While the method is well motivated theoretically and relative to prior work, the paper is occasionally difficult to follow. The bra-ket notation and operator formalism, while mathematically precise, may limit accessibility for a broader audience. The empirical approximations (Appendix E) are much clearer and could be introduced earlier to improve intuition. At times, the paper presents symbols and equations with insufficient context or motivation. Furthermore, the term dynamical richness is used somewhat loosely throughout; a dedicated subsection defining and contrasting rich vs lazy regimes would help.
- Unexplored temporal dynamics: The paper treats DLR at convergence or snapshot points, but since it is dynamical, a time-series analysis (e.g., how DLR evolves over training) would provide stronger evidence of what “rich” dynamics actually look like.

**Questions:**

- Typos: 049 "a dynamical richness (metric?) that", 373 "Our visualizes (visualizations)"
- How does the metric behave with different loss functions beyond MSE? The cross-entropy results are mentioned but not thoroughly analyzed.
- Metric sensitivity/ablation: More detail on estimation stability (sample size n for Nyström, dependence on width p, class count C) and hyperparameter sensitivity would help practitioners choose sampling parameters confidently.
- The batch normalization finding (last row of Table 3) is correlational. The paper doesn't establish that batch norm causes rich dynamics or explain the mechanism.
- The paper doesn't provide clear guidance on what intermediate DLR values mean.
- Interpretability link not quantified: The eigendecomposition visualization is qualitative; quantitative metrics (e.g., subspace alignment or effective rank) could have strengthened claims of interpretability.

---

> ### Author Response · Authors · 2025-11-19
>
> We appreciate the reviewer taking the time to review our paper and for providing detailed comments.
>
> ### Response to Weaknesses
>  * **Limited scope**: Isotropic target is a common assumption in NC literature, and handling anisotropy remains an active area of research. Please see 1 of the common comments.
>  * **Empirical validation**: Thank you for highlighting our experiments across various setups. Our models are comparable in scale to those used in studies of Neural Collapse (e.g. [1-4] in common comments) and significantly larger than those typically examined in lazy/rich dynamics (e.g. [5-7] in common comments). As lazy/rich dynamics remain a largely theory-driven area, we view our experiments not as a limitation but as an extensive empirical validation within this subfield.
>  * **Last-layer focus**: We added an appendix regarding intermediate layers. Please see 2 of our common comments.
>  * **Limited theoretical analysis**: The connection between richness and low-rank bias is well studied in the literature. See for example, Saxe et al. (2014) and Mixon et al. (2020), or Nam et al. (2025) for more gentle introduction.
>  * **Comparisons could be broader**: We did not use any scaled-down NTK comparisons or any NTK in our paper (Please see line 161). Assuming the reviewer is referring to $S_{init}$, tables 1 and 2 show that comparison to the initial last layer kernel ($S_{init}$) can have limitations. Based on our results on $S_{init}$ and the computational cost of NTK, we believe analysis on NTK has diminishing returns. Instead of NTK, we focused on theoretically well-studied lazy/rich transitions: Target downscaling (Chizat et al. 2019) in Table 2 (Fig. 11) and grokking (Kumar et al. 2024, Nam et al. 2025) in Table 3 (Fig. 12).
>  * **Clarity**: Thank you for taking the time to read our Appendix E. While we currently lack the space to incorporate Appendix E in the main text, we added more links to the relevant appendices (e.g., Appendix B when bra-ket notation is introduced a second time) to aid the readers. Please see 3 of the common comments.
>  * **Unexplored temporal dynamics**: We do have results on the temporal evolution (e.g. Figures 5, 14, and 17); the rich regime finds the low-rank structure first and increases its alignments in proportion to the feature intensity. Lazy regimes, by definition, do not have any change in their features. We again emphasize that our focus is not on understanding the details of rich dynamics, but to propose a metric that efficiently identifies lazy/rich regimes.
>
> ### Response to Questions
>  * **Typos**: Thank you for pointing out the typos. They are corrected in the revised version.
>  * **CE loss**: We expanded our analysis in Fig. 17. Fig. 17 shows dynamics similar to those under MSE loss, with the low-rank bias preserved but slightly stronger anisotropy in the top-$C$ features as CE loss is more punishing for misclassifications.  As our focus is on MSE, the standard choice in kernel and lazy/rich dynamics for its analytical tractability, the CE results are provided only as a proof of concept.
>  * **Metric sensitivity/ablation**: In the revised version, we added Appendix J, including the Nystrom method and the required sample size for large width. Please see 3 of our common comments.
>  * **Batch normalization**: We conducted a controlled experiment where all conditions (e.g., learning rate, initialization) were identical except for the use of batch normalization. Within this setup, the observed effect on richness can be considered causal. The underlying mechanism, however, lies beyond the scope of this paper and merits separate study, which we are currently pursuing. Briefly, the rescaling factor in batch normalization helps correct layer imbalance caused by large initializations: a phenomenon that can be analyzed using theoretical lazy/rich models (Kunin et al. 2025 , Domine et al. 2025, or Nam et al. 2025).
>  * **Intermediate $D_{LR}$**: Lazy and rich regimes are two ends of a smooth spectrum. Intermediate values of $D_{LR}$ mean the model is not on both ends but somewhere in the middle.
>  * **Interpretability link not quantified**: Our visualization contains three vector-valued metrics, offering a more qualitative perspective than the scalar $D_{LR}$. While metrics (ii) and (iii) may be represented as two scalar metrics (subspace alignment and effective rank), we adopt a full decomposition approach to complement the scalar metric $D_{LR}$.
>
>
>
> ### References:
>
> Saxe et al., (2014); Mixon et al., (2020); Nam et al., (2025); Chizat et al., (2019); Kumar et al., (2024); Baker & Taylor, (1979), Domine et al., (2025) are cited in the original submission manuscript.

---

### Official Review · Reviewer_GFrJ · 2025-10-30

**Soundness:** 4
**Presentation:** 4
**Contribution:** 4
**Rating:** 8
**Confidence:** 2

**Summary:**

In this paper, the authors develop the ***low-rank measure*** \$\mathcal{D}_{LR}\$ to measure the richness of the representations learned by a neural network, independently of its performance. Using it, and comparing it with the metrics from the existing literature, they can efficiently measure whether a model is in the *lazy* or *rich* regime.

The authors first introduce \$\mathcal{D}_{LR}\$ theoretically, basing it on the concept of low-rank bias where in rich dynamics, the rank of the features learned by the model before the final linear layer should be similar to the dimensionality of the output, indicating well-separated classes. The metric itself is simple, as it is based on the CKA between the representations after the penultimate and the ultimate layer. It is therefore also inexpensive to compute.

They then compare their metric with other ones from the rich dynamics literature. Across two different test cases, they demonstrate it better illustrates rich-vs-lazy dynamics than other metrics. They further test their metric on realistic use-cases, showing with different examples how it represents richness, and where it is related (or not) to model performance.

The last part of the paper focuses on visualisations to explain shifts in the learning dynamics of neural networks. The authors separate three different metrics through eigendecomposition of kernel: cumulative quality, cumulative utilisation and relative eigenvalue. They demonstrate how they relate to rich dynamics.

**Strengths:**

I think that this paper is a strong paper, which proposes a new metric for dynamic richness and demonstrates how it works better than other examples from recent literature.

S1: The *low-rank metric* is simple to understand and inexpensive to compute. The authors give good intuition on how it works, why it works, and bounds on computation complexity.

S2: Theoretical foundations for the *low-rank metric* look solid. Although the authors make some assumptions to make calculations tractable, formulations are very general and encompass a large portion of neural networks.

S3: Empirical evidence covers the validity of the metric, comparison to existing other metrics from the literature, and practical use cases. The authors further provide statistical significance results. I think it strongly supports the main claims of the paper. Furthermore, the authors propose visualisation methods for increased interpretability of their results.

S4: The literature review seems extensive, and most claims are backed through proofs or citations. This work is well contextualised within the existing literature.

S5: I strongly commend the authors’ work on making their paper clear and easy to read. Writing and presentation are of high quality. It allows readers to more easily understand the theoretical framework around the development of the *low-rank metric*.

**Weaknesses:**

I have not found major weaknesses in this paper. The following points are either minor or nitpicks.

W1: I feel like the authors should find a clearer name and/or an acronym to designate the *low-rank metric*, which is often designated as “the metric” or as its mathematical notation \$\mathcal{D}_{LR}\$ throughout the paper. Such a name/acronym would make it easier to designate and reference in text, discussions and future work that will rely on it.

**Questions:**

Q1: It is not immediately clear to me how exactly the metrics introduced in Section 5 relate to the *low-rank metric*. They are derived from \$\mathcal{T}\$’s eigendecomposition and the learned/target functions, but I have difficulties directly linking them to \$\mathcal{D}_{LR}\$.

Q2: The authors mention two main limitations: focus on last-layer dynamics, and validity constrained to orthogonal and isotropic target functions. Regarding the second point: the authors mentions that “while this covers most classification tasks, a more general setup would be preferable”. Could the authors please explain in what cases their method would not work, and/or would not be theoretically justified, and give some examples? Also, are there practical scenarios where this limitation could lead to misleading conclusions if the metric is applied without caution? If relevant, including such details in the paper could help practitioners understand when to avoid or adapt the use of this metric in their work.

---

> ### Author Response · Authors · 2025-11-19
>
> We thank the reviewer for the positive assessment, especially on clarity, and for constructive feedback.
>
> ### Response to Weaknesses
> **W1.** We appreciate your suggestion to find a clearer name for our low-rank metric. Currently, we are inclined to name it Minimum Projection Alignment to highlight the alignment between $T$ ($\approx$ features before the last layer) and Minimum projection operator $T_{MP}$ ($\approx$ the output). We are happy to take suggestions. We have not yet made the name changes in the current revision (except the Section title 3), but we will apply the changes once the name is fixed.
>
> ### Response to Questions
> **Q1.** The metric implicitly assumes $p > C$, meaning that the input dimension to the last layer is much larger than its output dimension. The information contained in the features before the last layer — at most $p$-dimensional random variable — is encoded in $T$ as an operator, while the output after the last layer — a $C$-dimensional random variable — is (loosely speaking) encoded in $T_{MP}$. The CKA term acts as an inner product between these two operators. A large $CKA(T, T_{MP})$ (i.e. small $D_{LR}$) indicates strong alignment between $T$ and $T_{MP}$, implying that the features before the last layer are effectively confined to a $C$ out of $p$-dimensional function space: a lower-rank bias. We have clarified this point further in Section 3.1. Thank you for your comment.
>
> **Q2.** Assume an extremely imbalanced binary classification where one class appears in 99.9% of the samples. A reasonable low-rank representation of this function would be to send a one-dimensional signal (i.e. $T$ is rank 1), which will predict the first class mindlessly. $T_{MP}$, however, assumes an isotropic target and will have rank 2. The alignment $CKA(T,T_{MP})$ would be small even when $T$ has found a low-rank representation. The centering in CKA will even lower the alignment.
>
> In short, $T_{MP}$, like neural collapse, is founded on the symmetry, and the current formalism requires a deeper understanding to correctly reflect the low-rank bias under such anisotropy. Please see 1 of our common comments.

---

> > ### Comment · Reviewer_GFrJ · 2025-11-20
> > **Rebuttal acknowledgement**
> >
> > I would like to thank the authors for their clear, concise and timely rebuttal.
> >
> > **W1**: This is a very minor point which I am happy to leave at the authors' discretion.
> >
> > **Q1**: I thank the authors for their explanation, and the addition of a few pieces of text in the manuscript which have helped me understand this point. I consider this question properly answered.
> >
> > **Q2**: I thank the authors for this example, which also helps me build intuition. I also consider this question to be answered.
> >
> > Following up, I have also read through the other reviews. I think the other reviewers raise a number of interesting points.
> >
> > **Limitation to orthogonal \& isotropic targets**: I think that this limitation is properly addressed and made explicit in the related Section, as mentioned in my Q2. Furthermore, I think the addition of an extended discussion in Appendix D.4 is good, and contextualises this limitation within the existing NC literature well. While this limitation hinders the applicability of the authors' work in a number of cases, I am convinced that generalising this work to non-isotopic targets can be left to future work, given the complexity of that task.
> >
> > **Analysis of intermediate layers**: I appreciate the addition of Appendix I to discuss and explore this limitation. The results in that Appendix support the choice of working at the last layer. The authors further highlight some limitations in studying features from intermediate layers. I think that the authors have correctly highlighted this limitation, and the addition of a discussion and exploration further strengthen the submission.
> >
> > **Clarity and accessibility**: Furthering my point S5, I think the authors have done excellent work in making their paper both general and accessible. I think it is a good decision to keep the main text formal and precise, while providing more intuition and reading helpers as appendices. The small improvements made in the revision are good and further help readability and clarity.
> >
> > In conclusion, I am still convinced this is an impactful and solid paper submission. Reviewers have raised multiple good points, which were partly already acknowledged in the *Limitations* of the paper, and have been correctly addressed in the rebuttal. The revision of the submission improves its clarity and furthers the discussion of these limitations. For these reasons, I am happy to not change my rating and still recommend an accept.

---

> > > ### Author Response · Authors · 2025-11-20
> > >
> > > We sincerely thank the reviewer for the careful reading and active engagement in the review process. We are grateful of the time reviewer invested in reviewing our paper.

---

### Author Response · Authors · 2025-11-19
**Common Comments (first half)**

We thank the reviewers for taking the time to review our paper and for highlighting three aspects of our work: the assumption of isotropic targets, the role of intermediate layers, and accessibility to a broader audience. In response, we have **(i)** added a paragraph in **Appendix D** explaining the standards in neural collapse (NC) literature and describing the challenges posed by anisotropic targets, **(ii)** included a new **Appendix I** (Intermediate Layer Features) reporting additional experiments and analysis on intermediate layers, and **(iii)** added a new **Appendix J** (Accuracy of Approximation Methods) that introduces the Nyström method and other approximations, along with additional references to the relevant appendices. We address each of these points in detail below.

We also note that the first two aspects, as noted by the reviewers, are explicitly listed as limitations in the main text, and that the original submission already included an extensive pedagogical appendix with a glossary, technical background, and a gentle introduction to kernel methods and neural collapse.

Finally, we note that all modifications and additions in the revised version are highlighted in red.

# 1. Non-isotropic target
As stated at the beginning of Section 2 and in the limitation paragraph, our paper’s scope is on isotropic outputs. We emphasize that while this is a limitation, it aligns with the prevailing standards in the NC literature [1-4], and is a much milder assumption compared to the standards in the literature on lazy/rich dynamics [5-7]. Additionally, handling anisotropy is an open challenge in the NC literature. We added this discussion in a new Appendix D.4.

**In Neural Collapse**: Neural collapse requires a slightly stronger assumption — balanced classification and the terminal phase of training. For example, neural collapse cannot handle scalar-output regression, which our formalism can handle (see Appendix D.3). Because simplex ETF structures rely significantly on the symmetry, proposing a plausible extension for imbalanced tasks is an active area of research on its own [8-12]. Additionally, it suffers from the minority collapse where multiple classes collapse to a single vector for a finite dataset [8]. This becomes particularly a problem for a long-tailed target (e.g., natural language dataset), which is also an active area of research [12].

**In lazy/rich dynamics**: Theories from lazy/rich dynamics prove less useful as they often require even less practical constraints (e.g. isotropic input, infinitesimal initialization, 2-layer linear networks). Removing the constraints (e.g., isotropic input) remains challenging even for two-layer linear networks [13], often requiring conjectures about the underlying feature learning mechanism.


# 2. Analysis of intermediate layers
In the revised version, we added Appendix I to examine intermediate layers. Extending our method to intermediate layers is straightforward: simply replace the last-layer features $\phi(x)$ with those from an intermediate layer when defining $\mathcal{T}$ (Eq. 58). However, such an extension presents three major caveats: limited theoretical grounding, higher computational cost, and reduced empirical utility for our goal.

**Theoretical limitation**: Our method relies on linear correlations (or kernel methods). For the last layer, features are linearly transformed to the output, validating this approach; this justification weakens once nonlinear transformations intervene, as in intermediate layers.

**Computational cost**: Our metric scales quadratically with the width. The last layer is typically narrow ($\sim 10^3$), whereas intermediate layers—especially in CNNs—are much wider (Fig. 22), making the computation substantially more expensive.

**Empirical limitation**: Although we observe phenomena such as a depth-wise transition from many less-relevant features to fewer more-relevant ones (Fig. 18) and minimal learning in earlier layers under the lazy regime (Fig. 20), these observations do not provide additional discriminatory power for lazy vs. rich dynamics beyond what the last layer already offers. While potentially useful for future studies of rich-regime representations, intermediate-layer analysis is less suitable for efficiently detecting lazy/rich dynamics — **our primary goal**.

In Fig. 21, we experimented with greedy layerwise training [14], in which each layer is trained with an auxiliary classifier, frozen, and followed by training the next layer. Under this procedure, each trained layer displays dynamics similar to the penultimate layer, and feature collapse emerges earlier than under full backpropagation, accompanied by reduced overall performance. Although limited, this experiment suggests that the last two layers exhibit the strongest low-rank bias, further supporting our focus on the last layer.

---

> ### Author Response · Authors · 2025-11-19
> **Common Comments (second half)**
>
> # 3. Accessibility of Technical Content for a broader audience
> We deeply acknowledge the burden of facing new notations or less familiar literature. In the original submission, we provided a glossary in Appendix A; explanation of technical terms such as bra-ket notation and vector space formed by random variables in Appendix B; gentle introduction of kernel (operator) formalism with examples in Appendix C; introduction of Neural collapse and its connection to our metric in Appendix D; approximation methods of our visualization in Appendix E;  visualization of each cases of Table 3 in Appendix F; and related works on visualization in Appendix G. We believe the only missing review is on the lazy/rich dynamics, which was omitted as an introductory literature already exists [15].
>
> In the revised version, we added further cross-references to relevant appendices to help guide readers. We also added Appendix J, which introduces the Nyström method, a widely used approximation technique [16], with an example. We analyzed the number of samples required for reliable approximation in very wide layers, and a technique to handle inconsistency from finite sample size.
>
> We again thank all reviewers for their engagement in the reviewing process and for their comments.
>
> ### References:
>  * [1]* Papayan et al. 2020, Prevalence of neural collapse during the terminal phase of deep learning training.
>  * [2]* Mixon et al. 2020, Neural collapse with unconstrained features.
>  * [3] Han et al. 2021, Neural Collapse Under MSE Loss
>  * [4] Jacot et al. 2025, Wide Neural Networks Trained with Weight Decay Provably Exhibit Neural Collapse
>  * [5]* Saxe et al. 2014, Exact solutions to the nonlinear dynamics of learning in deep linear neural networks.
>  * [6]* Braun et al. 2022, Exact learning dynamics of deep linear networks with prior knowledge.
>  * [7]* Kunin et al. 2024, Get rich quick: exact solutions reveal how unbalanced initializations promote rapid feature learning.
>  * [8] Fang et al. 2021, Exploring deep neural networks via layer-peeled model: Minority collapse in imbalanced training.
>  * [9] Yang et al. 2022, Inducing Neural Collapse in Imbalanced Learning: Do We Really Need a Learnable Classifier at the End of Deep Neural Network?
>  * [10] Hong et al. 2024, Neural Collapse for Unconstrained Feature Model under Imbalanced Settings.
>  * [11] Yan et al. 2024, Neural Collapse to Multiple Centers for Imbalanced Data.
>  * [12] Gao et al. 2024, Distribution Alignment Optimization through Neural Collapse for Long-Tailed Recognition. 2024.
>  * [13] Kunin et al. 2025, Alternating Gradient Flows: A Theory of Feature Learning in Two-layer Neural Networks
>  * [14] Ivakhnenko and Lapa. 1965, Cybernetic Predicting Devices.
>  * [15]* Nam et al. 2025, Position: Solve layerwise linear models first to understand neural dynamical phenomena (neural collapse, emergence, lazy/rich regime, and grokking).
>  * [16]* Baker Taylor 1979, The numerical treatment of integral equations.
>
> Starred (*) papers indicate that the references exist in the original submission.

---

### Author Response · Authors · 2025-11-30

Dear AC,

First, we would like to express our regret regarding the recent incident and our understanding of the additional burden it has placed on the AC. We hope the community will recover promptly. Below, we provide a summary of our discussion.

The main topics raised were the two limitations mentioned in our limitations section, as well as the accessibility of the technical content. In response, we added Appendices D.4, I, and J. Further details are provided in our common comments (Nov. 19) and in the individual rebuttals.

Reviewers GFrJ (Nov. 20) and cwga (Nov. 28) maintained their score of 8, supporting acceptance. Reviewer Czxk (Nov. 21) raised their score from 4 to 6, noting that the contribution is non-trivial, while also expressing concerns that some points, though better addressed, remain open. Reviewer ztAm did not respond before the rollback.

Although we believe a few points raised did not fundamentally undermine the contribution, we nonetheless found the discussion constructive and thoughtful. It enabled us to further refine the manuscript through additional appendices and various improvements (including typo corrections), and we sincerely thank all reviewers/AC/SAC for their time and engagement in the review and rebuttal process.

---

### Meta-Review · Area_Chair_AT86 · 2026-01-08

**Summary:**

The paper investigates the quantification of neural network learning dynamics through the lens of feature space. The authors propose a computationally efficient low-rank measure to analyze the alignment between the features and the function space. By comparing their metric with existing measures in the literature through both analysis and empirical evaluation, the authors demonstrate that their proposed measure effectively characterizes shifts in learning dynamics.

Most initial concerns have been addressed during the rebuttal phase. While the limitation to isotropic targets remains a point of discussion, the reviewers generally agree that the paper makes a solid contribution, and view the proposed measure as a valuable tool for characterizing feature evolution and distinguishing between lazy and rich learning regimes.

**Reviewer Concerns:**

Main concerns include the fact that the analysis was limited to last-hidden-layer representations and restricted to orthogonal and isotropic targets. To address the former, the authors provided additional experiments on intermediate layers during the rebuttal. These results demonstrate phenomena similar to those observed in the penultimate layer, which largely resolves the concern regarding layer-wise generalizability. The second concern regarding target constraints remains, although the authors argue that this limitation is standard in existing literature on Neural Collapse and lazy/rich dynamics. Other concerns appear to have been largely addressed, with the exception of the critique regarding the small scale of the experiments (for which the authors do not provide additional experiments, but noted that the experiments as an extensive empirical validation within this subfield).

**Reviewer Scores:**

The paper originally received scores of 8, 8, 4, and 4. Both reviewers who gave an 8 indicated that their concerns have been addressed and maintained their positive stance. Reviewer Czxk noted that their concerns were partially resolved and subsequently raised their score from 4 to 6. Had Reviewer ztAm been able to participate fully in the discussion, it is possible that they would adjust the score given the new experimental results extending the analysis to intermediate layers. However, it is also plausible that the reviewer would have maintained their original score, as the concerns regarding the small-scale datasets and the restriction to orthogonal and isotropic targets remain.

---

### Decision · Program_Chairs · 2026-01-26

Accept (Poster)